# Identification of ephrin-A1–EphA2 signalling as a potential target for fracture prevention

Sofia Movérare-Skrtic [1,4] ✉, Maria Nethander [1,4], Lei Li[1], Nelson Tsz Long Chu [1], Ostap Dregval [1], Xin Tian[1], Karin H. Nilsson [1], Petra Henning[1], Ulf H. Lerner[1], Andrei S. Chagin [1,2] & Claes Ohlsson [1,3] ✉

Osteoporotic fractures are a major global health burden. To uncover potential targets for fracture prevention, we use a proteome-wide Mendelian randomization (MR) approach combined with colocalization. Here we show that nine circulating proteins associate with forearm fracture risk, including sclerostin and osteoprotegerin targeted by existing osteoporosis treatments, and three other known bone-related proteins, providing proof of concept for our MRpipeline. Notably, we identify ephrin-A1 as a novel protective factor against fractures, a membrane-linked protein partly released into circulation that binds its high-affinity receptor EphA2 on osteoblasts. Experimental models and genetic analyses indicate that ephrin-A1 increases bone mineral density, supporting a mechanism by which this pathway may mediate fracture protection. Spatial expression analysis with the innovative 3D DeepBone technique suggests ephrin-A1 on endothelial cells interacts with EphA2 on adjacent osteoblasts at the bone surface. These findings position ephrin-A1–EphA2 signalling as a therapeutic target to strengthen bone and reduce fracture risk.

Osteoporosis, characterised by reduced bone mass and deteriorated bone architecture, is a common complex disease. The increased bone fragility leads to an increased fracture risk, with every 1 in 2 women and 1 in 4 men experiencing an osteoporotic fracture during their lifetime[1,2]. Simple and effective fracture preventive strategies that could be widely implemented are crucial, especially since current osteoporosis medications only have indications for treating established osteoporosis[3]. In addition, prolonged treatment with current osteoporosis medications has been associated with increased risk of serious but rare adverse events[4,5]. It is estimated that as many as 14 million women in the EU who were eligible for treatment are untreated, partly due to increasing concern regarding side effects associated with currently available osteoporosis drugs[6].

Circulating proteins are targets for treatments of a wide variety of diseases, and increased understanding of the role of circulating proteins for fracture risk may identify novel biological pathways and therapeutic targets for fracture prevention[7–9]. Combining human genetics with high-throughput, population-scale circulating proteomics is a recently developed methodology to bridge the gap between the human genome and human diseases. This methodology has been facilitated by the characterisation of the genetic architectures of the circulating proteome, providing genetic instruments for a large proportion of the circulating proteins[7,8]. Genetic instruments that determine circulating levels of proteins are known as protein-quantitative trait loci (pQTLs). PQTLs, particularly those lying in or near a protein's cognate gene (referred to as cis-pQTLs), can be leveraged to identify candidate aetiological proteins for fracture risk through Mendelian randomisation (MR) analyses[10]. MR is a method of causal inference that uses genetic variants as instrumental variables to test the role of exposures in disease outcomes. MR should be

[1]Department of Internal Medicine and Clinical Nutrition, Institute of Medicine, Sahlgrenska Osteoporosis Centre, Centre for Bone and Arthritis Research at the Sahlgrenska Academy, University of Gothenburg, Gothenburg, Sweden. [2]Science for Life Laboratory, Department of Internal Medicine and Clinical Nutrition, Institute of Medicine, Sahlgrenska Academy, University of Gothenburg, Gothenburg, Sweden. [3]Unit of Clinical Pharmacology, Department of Pharmaceuticals, Sahlgrenska University Hospital, Gothenburg, Region Västra Götaland, Sweden. [4]These authors contributed equally: Sofia Movérare-Skrtic, Maria Nethander. ✉e-mail: sofia.skrtic@gu.se; claes.ohlsson@medic.gu.se

complemented with colocalization analyses to exclude confounding by linkage disequilibrium (LD)[11].

The aim of the present study was to identify novel biological pathways and targets for fracture prevention. To this end, we applied an integrated cis-pQTL MR and colocalization pipeline, followed by mechanistic studies of the most promising novel proteins (Fig. 1a). For the mechanistic studies, we employed gene-targeted mouse models combined with detailed transcriptional analyses of bone tissue, utilising single-cell RNA sequencing, RNAscope, and 3D spatial expression analyses. For the MR analyses, we used valid genetic instruments for 1615 circulating proteins, derived from UK Biobank, as exposures and summary statistics from the largest available fractures genome-wide association study (GWAS; > 50,000 forearm fractures) for the outcome analyses[7,12]. First, we assessed the performance of the developed

MR pipeline by determining if established, approved osteoporosis treatments were successfully identified. Our identification of two proteins directly involved in the mechanisms underlying two widely used osteoporosis treatments serves as proof-of-concept of the efficacy and accuracy of our developed MR pipeline, suggesting that other novel proteins identified by this MR pipeline are likely to be biologically and clinically relevant.

## Results

### Mendelian randomisation identified nine circulating proteins causally associated with fractures

To identify proteins causally associated with fractures, we used recently developed genetic instruments for circulating proteins[7] in MR of fractures, using the largest available fractures GWAS for the outcome

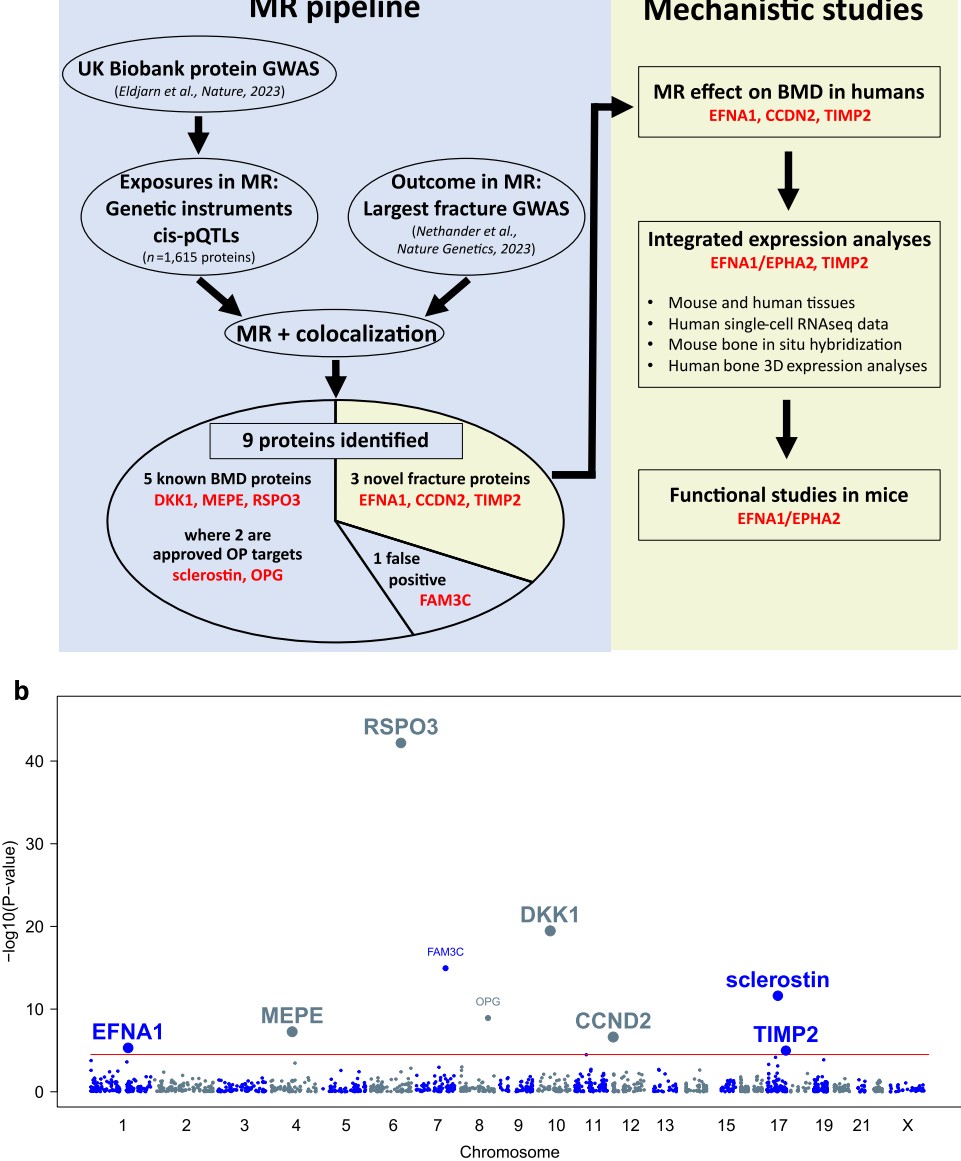

**Fig. 1 | Study design and genetic associations of circulating proteins with fracture risk. a** Overview of the design of the present translational study. **b** Association of genetically predicted protein concentrations (*n* = 1615 proteins) with fracture risk presented as a Manhattan plot. The position is given by cis-pQTL coordinate (chromosome and base-pair position) labelled with its association with fracture risk and the highest colocalization probability from single or conditional iterative methods (PP.H4). Unadjusted *P*-values are from MR analysis using a two-sided z-test. Proteins highlighted as large font in bold are those with evidence of a shared causal locus (PP.H4 > 0.8), with point size reflecting PP.H4 magnitude, which can vary between 0 and 1. The red horizontal line marks the Bonferroni-corrected threshold ($P = 3.1 \times 10^{-5}$); associations below this threshold were not subjected to colocalization analysis. OP osteoporosis; BMD bone mineral density; GWAS genome-wide association study; MR Mendelian randomisation; pQTL protein quantitative trait locus.

**Table 1 | Identification of causal circulating proteins for forearm fractures using an MR pipeline**

| Gene | Protein | Beta | SE | P | Colocalization | Previous MR evidence for fractures in humans | Known to associate with BMD in humans | Approved osteoporosis drug |
|------|---------|------|----|----|----------------|----------------------------------------------|---------------------------------------|----------------------------|
| RSPO3 | R-spondin 3 | −0.58 | 0.04 | 6.5E-43 | Yes | Yes | Yes | – |
| DKK1 | Dickkopf-related protein 1 | 0.44 | 0.05 | 3.4E-20 | Yes | – | Yes | – |
| FAM3C | FAM3C | −0.60 | 0.07 | 1.1E-15 | – | – | – | – |
| SOST | Sclerostin | 0.76 | 0.11 | 2.4E-12 | Yes | Yes | Yes | Yes |
| TNFRSF11B | Tumour necrosis factor receptor superfamily member 11B | −0.19 | 0.03 | 1.2E-09 | – | – | Yes | Yes |
| MEPE | Matrix extracellular phosphoglycoprotein | −0.26 | 0.05 | 5.4E-08 | Yes | – | Yes | – |
| CCND2 | Cyclin D2 | 0.66 | 0.13 | 2.4E-07 | Yes | – | – | – |
| EFNA1 | Ephrin-A1 | −0.11 | 0.02 | 4.9E-06 | Yes | – | – | – |
| TIMP2 | Metalloproteinase inhibitor 2 | 0.34 | 0.08 | 1.0E-05 | Yes | – | Yes | – |

Unadjusted P values are from two-sided z-tests. SE standard error of Beta; MR Mendelian randomisation; BMD bone mineral density; Tumour necrosis factor receptor superfamily member 11B, osteoprotegerin (OPG).

analyses (forearm fractures[12]). As exposures in the MR analysis, we used genetic variants that determine circulating protein levels, known as protein-quantitative trait loci (pQTLs). We only used cis-pQTLs and excluded protein altering variants. We identified 1,615 circulating proteins (Supplementary Data 1) with valid genetic instruments (Fig. 1a). The MR pipeline developed herein (Fig. 1a) identified nine circulating proteins causally linked to forearm fractures (Fig. 1b, Table 1 and Supplementary Data 1; Bonferroni correction for 1,615 proteins, $P < 3.1 \times 10^{-5}$). Strong evidence for colocalization was observed for seven of these proteins (PP.H4 > 0.80; Fig. 1b, Table 1 and Supplementary Data 2), demonstrating a shared causal variant in the identified protein locus affecting both the circulating protein levels and the risk of forearm fractures. Two of the identified proteins (sclerostin and osteoprotegerin) are directly involved in the different mechanisms targeted by two approved osteoporosis treatments (romosozumab and denosumab)[13–15] and three other proteins (DKK1, RSPO3, and MEPE) are well-established bone-related proteins (Fig. 1)[16–19]. These findings serve as proof-of-concept that our developed MR pipeline effectively identifies clinically relevant causal proteins for fractures.

Our MR analyses indicated that genetically predicted FAM3C was associated with forearm fracture risk (Table 1). However, there was a modest correlation between the cis-SNP (rs138090420) of circulating FAM3C and the top known bone-related SNP in the neighbouring WNT16 locus (rs2908007)[12], suggesting that these two signals are not independent signals ($R^2 = 0.052$ and $D' = 1.00$). An association driven by LD is supported by the fact that the FAM3C protein signal was not colocalised with the fracture signal in this region (Table 1 and Supplementary Data 2). In addition, the significant association between the FAM3C signal (rs138090420) and forearm fractures was lost when adjusting for the top known bone-related SNP in the WNT16 locus (rs2908007; Supplementary Data 3)[12]. Collectively, these findings do not provide strong evidence that FAM3C is a WNT16-independent target for forearm fractures.

Interestingly, we identified cyclin D2 (CCND2), ephrin-A1 (EFNA1), and metalloproteinase inhibitor 2 (TIMP2) to be causally associated with forearm fractures (Fig. 1b and Table 1). The genetic signals of these three proteins, not previously shown to be linked to fractures, were all colocalised with corresponding genetic signals for forearm fractures (Table 1 and Supplementary Data 2). Thus, our MR pipeline identified five well-known therapeutic targets for osteoporosis, and three novel potential targets, which we further explored below.

## Bone mineral density as a possible mediator of the effects for the identified novel proteins on forearm fracture risk

We determined if any of the three identified novel candidate fracture proteins (CCDN2, TIMP2, EFNA1) were causally associated with

estimated BMD (eBMD) in the heel. The selection of eBMD as a BMD outcome was partly due to the availability of a large GWAS[20] enhancing the statistical power of the MR outcome analyses, but also because eBMD is a good predictor of trabecular BMD in the distal radius and forearm fractures[21]. All these three candidate fracture proteins were significantly causally associated with eBMD (Supplementary Data 4). EFNA1 and TIMP2, but not CCDN2, were also causally associated with total body BMD, when evaluated using a large total body DXA GWAS meta-analysis for the outcome analyses in MR (Supplementary Data 4)[22]. These findings suggest that these three proteins may exert some of their effects on fracture risk via effects on BMD.

CCND2, TIMP2, and EFNA1 as potential targets for reducing forearm fracture risk among the three potential novel targets identified for forearm fracture risk, CCND2 encodes cyclin D2, a key intracellular regulator of the G1–S cell cycle transition across multiple cell types. Due to its broad role in cell proliferation, CCND2 is unlikely to serve as a viable, specific therapeutic target for fracture prevention[23,24].

Interestingly, although TIMP2 is described to influence bone homoeostasis[20,23–27], our MR and colocalization analyses provide the first evidence linking TIMP2 to fracture risk. GTEx humans tissue panels revealed the highest TIMP2 expression in arterial tissues, but bone was not included in the dataset (Supplementary Fig. 1a). To address this, we analysed Timp2 mRNA expression in various mouse tissues and observed the highest levels in the hypothalamus and relatively high expression in diaphyseal cortical bone (Supplementary Fig. 1b, c). Further, using our recently developed single-cell RNA sequencing atlas of human bone marrow cells[28], which includes hematopoietic, osteolineage, chondrocytic, stromal, endothelial, and smooth muscle cell populations (Supplementary Fig. 1d), we found strong TIMP2 expression in osteoblast-lineage and bone marrow stromal cells (Supplementary Fig. 1e). These findings position TIMP2 as a compelling candidate for fracture-preventing therapy.

Most notably, our MR analyses uncovered that high circulating EFNA1 levels were causally linked to reduced forearm fracture risk and high BMD in humans. To our knowledge, EFNA1 has not previously been implicated in osteoporosis or fracture susceptibility. Given its potential as a novel therapeutic target (Table 1), below we conducted expression profiling and functional studies to characterise the EFNA1 signalling in bone, with a particular focus on ephrin type-A receptor 2 (EPHA2), a high-affinity receptor for EFNA1[29,30]. We hypothesised that EFNA1 may increase bone mass via activation of EphA2 in bone.

As EFNA1 was selected as a promising candidate for our further downstream analyses, we first conducted MR sensitivity analyses using alternative genetic instruments for EFNA1. The results from these MR sensitivity analyses, employing alternative genetic instruments for EFNA1, yielded effect estimates for the association between genetically

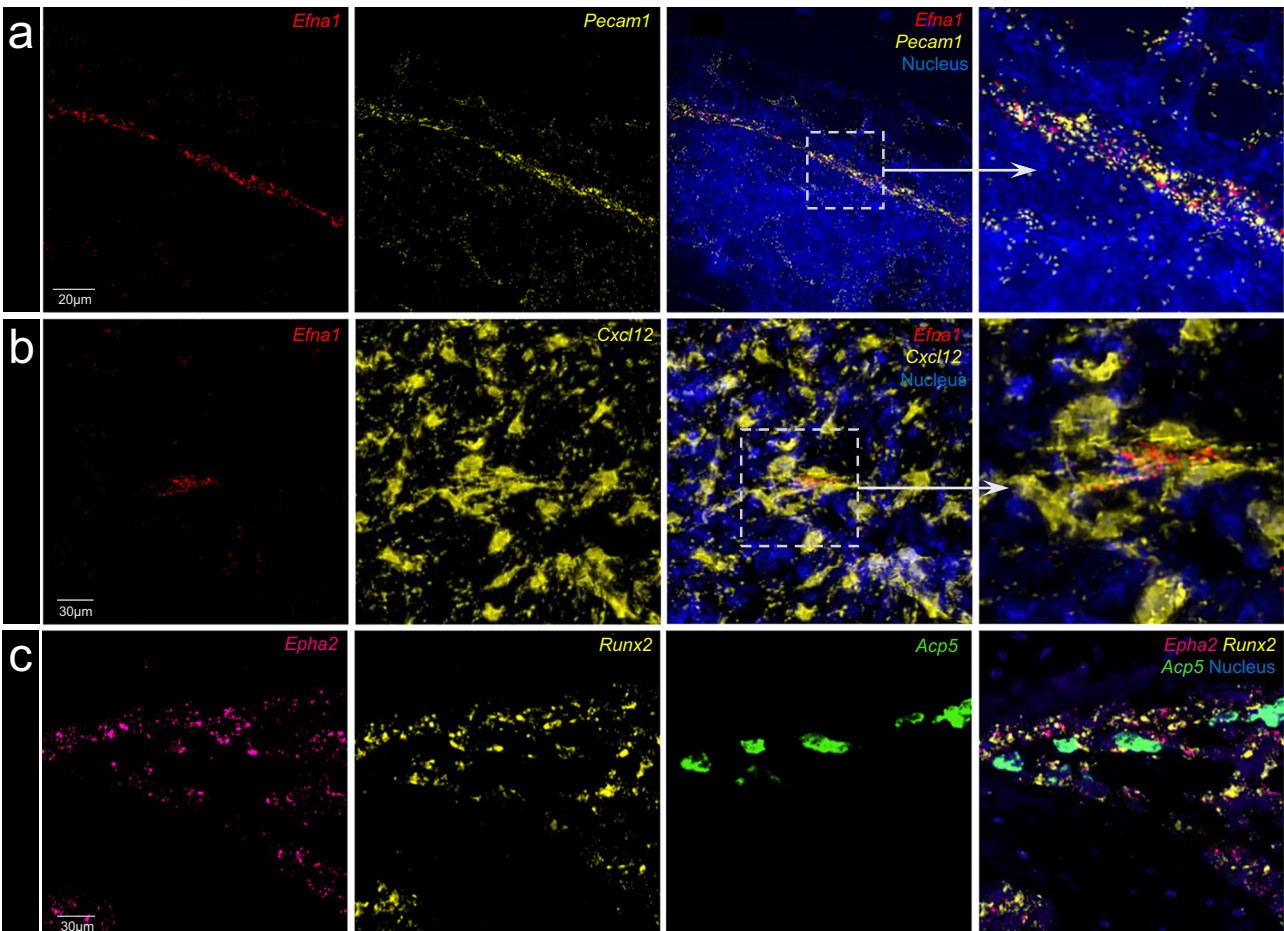

**Fig. 2 | Expressions of *Efna1* and its receptor *EphA2* mRNA in adult mouse long bone.** Multiplex fluorescence in situ hybridisation (FISH) combined with confocal imaging was used to visualise *Efna1*, *EphA2*, and associated transcripts in decalcified sections of adult mouse long bone. **a** Blood vessels in the trabecular region labelled with *Pecam1* (CD31) mRNA (yellow), a marker of vascular endothelial cells. *Efna1* mRNA (red) is detected in the same structures, and nuclei are counterstained with DAPI (blue). **b** Bone marrow regions showing partial colocalization of *Efna1* (red) with *Cxcl12* mRNA (yellow) in a subset of stromal cells, while other cells express only one of the two transcripts. **c** *EphA2* mRNA (purple), encoding a high-affinity receptor for Efna1, is observed in cells expressing *Runx2* mRNA (yellow), a transcription factor associated with osteoprogenitor cells, without overlap with *Acp5* mRNA (green), expressed by mature osteoclasts. All FISH staining experiments were performed on bone sections from six independent adult mice and repeated in a separate session, with consistent observations across all replicates. FISH fluorescence in situ hybridisation; DAPI 4′,6-diamidino-2-phenylindole.

predicted circulating EFNA1 levels and forearm fracture risk that were largely consistent with that reported in the present study (Supplementary Data 5).

In addition, we assessed the association between genetically predicted circulating levels of EFNA1 and risk of fracture at any skeletal site[20]. This analysis revealed a statistically significant association in the same direction as that observed for forearm fractures (Supplementary Data 5).

### Identification of ephrin-A1–EphA2 signalling as a target for fracture prevention

The human GTEx tissue panel showed that the highest *EFNA1* expression was observed in liver (Supplementary Fig. 2a). Within the liver, the highest expression of *EFNA1* was observed in endothelial cells and hepatocytes, as predicted by single-cell RNA sequencing data of human liver cells (Supplementary Fig. 3). Thus, it is possible that circulating EFNA1 is partly liver-derived, but it is also possible that the circulating levels of ephrin-A1 mainly reflect the local ephrin-A1 expression in the bone microenvironment. The highest *EPHA2* expression was observed in the esophagus (Supplementary Fig. 2b), but bone is not included in the human GTEx expression panel. Therefore, we analysed *Efna1* and *EphA2* expression in several mouse

tissues to determine relative expression in bone in comparison with other tissues (Supplementary Fig. 4). The highest *Efna1* expression was observed in colon and liver, partially resembling humans, but the expression was also substantial in both cortical and trabecular bone (Supplementary Fig. 4a, b). The expression of *EphA2* was among the highest in both cortical and trabecular bone (Supplementary Fig. 4c, d).

To provide further evidence for expression of ephrin-A1 and its high-affinity receptor *EphA2* in bone and to determine their cellular localisation within bone, we applied RNA-FISH on mouse and human bone tissue. RNAscope analyses of long bones in mice revealed high *Efna1* expression by *Pecam1* (coding for CD31)-positive endothelial cells in the vicinity of small blood vessels in bone (Fig. 2a). *Efna1* expression was also observed in some bone marrow stromal cells, identified by *Cxcl12* expression and characteristic morphology (Fig. 2b)[31], and *Runx2* positive osteoblast-lineage cells (Supplementary Fig. 5). *EphA2* was highly expressed by *Runx2*-positive osteoblasts on bone surfaces, but not by *Acp5*-positive osteoclasts (Fig. 2c). Next, we explored the single-cell RNA sequencing dataset of human bone marrow, which revealed high *EFNA1* expression in endothelial cells and bone marrow mesenchymal stromal cells, while *EPHA2* was highly expressed in osteoblast-lineage cells (Fig. 3a–c and

Supplementary Fig. 7c). When comparing the expression pattern of the different *EPHA* receptors in human bone marrow, the highest expression in osteoblast-lineage cells was observed for *EPHA2*, which has high affinity for *EFNA1*, and the second highest expression was observed for *EPHA3*, which has low affinity for *EFNA1* (Supplementary Figs. 6, 7c)[29].

The ligand-receptor interaction analyses between clusters in human bone marrow predicted an interaction between EFNA1 expressed by endothelial cells and its high-affinity receptor EPHA2[29] on osteoblasts (Supplementary Fig. 7d). For this analysis, we utilised the Rank Aggregate method from the LIANA package, which integrates and cross-validates predictions from multiple tools, including Cell-Chat, CellPhoneDB, CellSignalIR, Connectome, and NATMI[32].

To further explore the spatial distribution of *EFNA1* and *EPHA2*-expressing cells in human bone, we utilised our novel 3D DeepBone tissue-clearing technique combined with RNA-FISH[28]. Visualisation of *EFNA1* mRNA-expressing cells together with CD31-positive blood vessels revealed abundant *EFNA1* expression within the human bone marrow, including numerous *EFNA1*-positive cells in the vicinity of (Fig. 3d–f), and colocalising with, blood vessels (Supplementary Fig. 7e–g). Visualisation of *EPHA2* mRNA-expressing cells together with bone matrix showed a high number of *EPHA2*-expressing cells, especially in close proximity to the bone surface (Fig. 3g–i). Integrated 3D-spatial expression analyses showed that *EPHA2*-positive cells located along the bone surface are closer to *EFNA1*-expressing cells found along the blood vessels than to other *EFNA1*-expressing cells (Supplementary Fig. 8a–c).

This spatial distribution of *EPHA2*-expressing cells, combined with single-cell RNA sequencing data, suggests that osteolineage cells near the bone surface express this high-affinity receptor. Dual detection of *EPHA2* and the osteolineage marker RUNX2 (Supplementary Fig. 7b) revealed that *EPHA2* was expressed by $75.9 \pm 6.9\%$ of RUNX2-positive osteoblasts along the bone surface (Fig. 4a–d). Similar results were obtained with another marker for osteolineage cells, CD56 (NCAM1, Supplementary Fig. 7b), which showed $79.9 \pm 5.7\%$ of double-positive cells (Supplementary Fig. 8d–g). Integrated analysis of the spatial distribution of *EFNA1*-expressing cells showed that cells distributed along blood vessels are closer to RUNX2-positive osteoblasts on bone surfaces as compared with all other *EFNA1*-expressing cells (Fig. 4e–g). In fact, 19.1% of the vessel-associated *EFNA1*-positive cells were located within 20 µm distance to RUNX2-positive osteoblasts (Fig. 4h), suggesting a possibility of a direct interaction. These data suggest that cell-membrane linked ephrin-A1 on CD31-positive vascular endothelial cells may interact directly with the EPHA2 receptor on osteoblasts located at the bone surfaces.

The similarities observed between humans and mice in the expression patterns of *EFNA1* and *EPHA2* justified the use of mouse models for functional studies. We determined the consequences of the inactivation of *Efna1* and *EphA2* for bone mass. Analyses of total body BMD of data derived from the International Mouse Phenotyping Consortium database revealed that BMD was significantly reduced in both male and female *EphA2*[-/-] mice (Fig. 5a) and in female *Efna1*[-/-] mice (Fig. 5b). These functional data demonstrate that the ephrin-A1–EphA2 signalling pathway is crucial for maintaining bone homeostasis.

Finally, we investigated whether the expression of ephrin-A1 or its high-affinity receptor EphA2 in bone tissue are regulated across various osteoporosis models, including those with bone loss induced by inflammation, aging, high-dose vitamin A exposure, or oestrogen deficiency (Fig. 6). Inflammation-induced bone loss was associated with decreased expression of both ephrin-A1 and EphA2 in bone tissue (Fig. 6a). Bone loss associated with aging was linked to reduced levels of EphA2 and a non-significant trend toward decreased expression of ephrin-A1 in bone tissue (Fig. 6b). Bone loss induced by high-dose vitamin A was specifically associated with reduced expression of EphA2 in bone tissue (Fig. 6c). Ovariectomy-induced bone loss, which

models postmenopausal osteoporosis, was not associated with altered expression of ephrin-A1 or its receptor EphA2 in bone tissue (Fig. 6d). The anabolic bone response induced by mechanical loading, but not by intermittent administration of parathyroid hormone (PTH), was associated with increased expression of ephrin-A1 in bone tissue (Fig. 6e, f). The observed alterations in ephrin-A1 and EphA2 expression was dependent on the osteoporosis models, as well as on the type of anabolic stimuli evaluated. This suggests that ephrin-A1–EphA2 signalling in bone may constitute a mechanistically distinct pathway from those targeted by currently approved osteoporosis therapies. Nonetheless, further functional studies are needed to determine whether ephrin-A1–EphA2 signalling contributes to the observed bone effects in selected osteoporosis disease models or in the response to certain anabolic stimuli on bone mass.

## Discussion

Osteoporotic fractures are among the most common and costly of diseases[1,2]. To identify novel biological pathways and targets for fracture prevention, we conducted a proteome-wide MR and colocalization analysis followed by selected mechanistic studies. Using this strategy, we identified nine circulating proteins associated with forearm fractures, including seven with strong colocalization evidence. Two of the identified proteins (sclerostin and osteoprotegerin) are involved in the different mechanisms targeted by two widely used osteoporosis treatments[13–15] and three other proteins (DKK1, RSPO3, and MEPE) are well-established bone-related proteins[16–19]. As our MR analyses demonstrated that high circulating ephrin-A1 is causally associated with low forearm fracture risk and high BMD in humans, we hypothesised that ephrin-A1, via activation of its high-affinity receptor EphA2 in bone, increases bone mass. Detailed expression analyses using single-cell RNA sequencing and spatial expression analyses demonstrated that endothelial ephrin-A1 may interact directly with EphA2 on osteoblasts to increase BMD and thereby reduce fracture risk. This notion is supported by our functional observations that both *Efna1*[-/-] and *EphA2*[-/-] mice had reduced BMD. Collectively, we identified ephrin-A1–EphA2 signalling as a potential novel target for fracture prevention.

When establishing a new pipeline for the identification of novel drug targets, it is crucial to validate its effectiveness by determining whether already approved clinically relevant targets are identified. Importantly, two of the proteins identified by our MR pipeline are directly involved in the different mechanisms underlying two approved osteoporosis treatments with established anti-fracture data[13–15]. The Wnt signalling inhibitor sclerostin, identified in the present MR pipeline and supported by colocalization evidence, is the direct target of romosozumab, a monoclonal antibody against sclerostin[13,15]. The approved anti-fracture treatment denosumab[14] targets RANKL and shares functional similarities with circulating OPG, which was identified in the present MR pipeline, although without strong colocalization support. OPG is a secreted decoy receptor which binds to RANKL, thereby inhibiting osteoclast formation by preventing RANKL-RANK interaction. The direction of the causal effects of these two proteins were also as expected, as increased genetically predicted circulating sclerostin was casually associated with increased fracture risk, while increased circulating OPG was associated with reduced fracture risk. The identification of these two osteoporosis drug targets serves as proof-of-concept that our developed MR pipeline effectively identifies clinically relevant causal proteins for fractures. Thereby, other novel, fracture-related, proteins identified by this MR pipeline are also likely to be biologically and clinically relevant.

Higher genetically determined levels of another Wnt-signalling inhibitor Dickkopf-related protein 1 (DKK1) were also causally associated with increased fracture risk, supported by colocalization evidence. Although there is no approved osteoporosis treatment targeting DKK1, a recent clinical study demonstrates beneficial effects

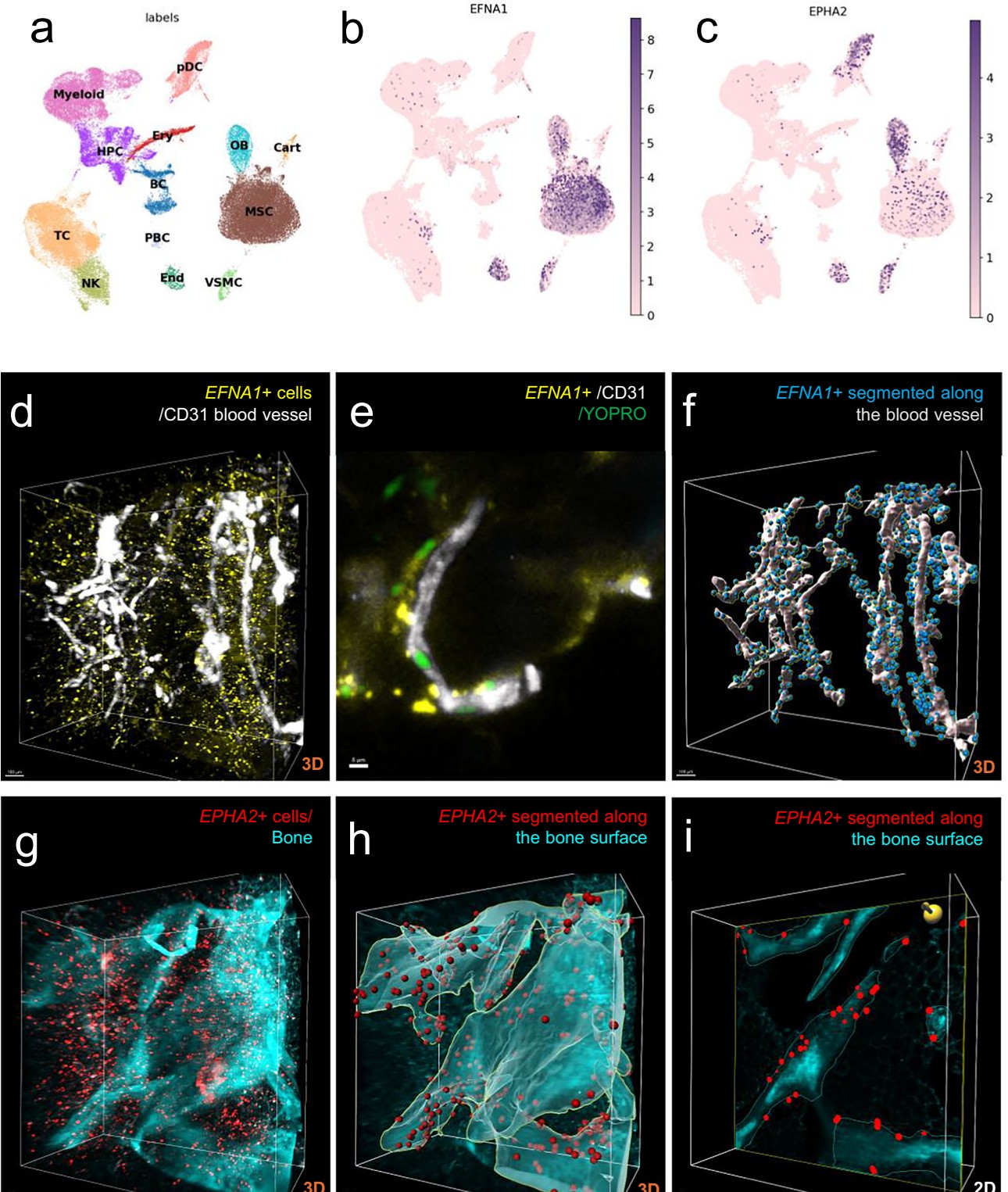

**Fig. 3 | Expression and localisation of EFNA1 and EPHA2 in human bone.**
**a** Embedding of the combined human bone marrow single-cell RNA sequencing atlas. Broad cell identities according to marker gene expression. BC, B cells; Cart, cartilage cells; End, endothelial cells; Ery, erythrocytes; HPC, hematopoietic progenitors; MSC, mesenchymal stromal cells; myeloid, myeloid cells; NK, natural killer cells; OB, osteolineage cells; PBC, plasma B-cells; TC, T cells; VSMC, vascular smooth muscle cells; pDC, plasmacytoid dendritic cells. **b**, **c** Single-cell gene expression levels of *EFNA1* (**b**) and *EPHA2* (**c**), normalised by sequencing depth. **d** *EFNA1* mRNA expression (pseudocolored yellow) is shown in relation to CD31-positive blood vessels (pseudocolored white) in cleared human bone. A maximum

intensity projection of the 3D scan is displayed. **e** Corresponding optical section from the same scan. YOPRO-stained nuclei (pseudocolored green) were excluded from panel (**d**) for visual clarity. **f** Segmentation of blood vessels with *EFNA1*-positive cells located within 20 μm of vessels (defined as vessel-associated cells).
**g** *EPHA2* mRNA expression (pseudocolored red) in cleared human bone, shown together with bone matrix visualised via autofluorescence at 405 nm (pseudocolored cyan). A maximum intensity projection of the 3D scan is shown.
**h**, **i** Segmentation of the bone surface and *EPHA2*-positive cells located within 20 μm of this surface. **h** Maximum intensity projection of the segmented 3D scan. **i** Optical section from the scan in (**h**).

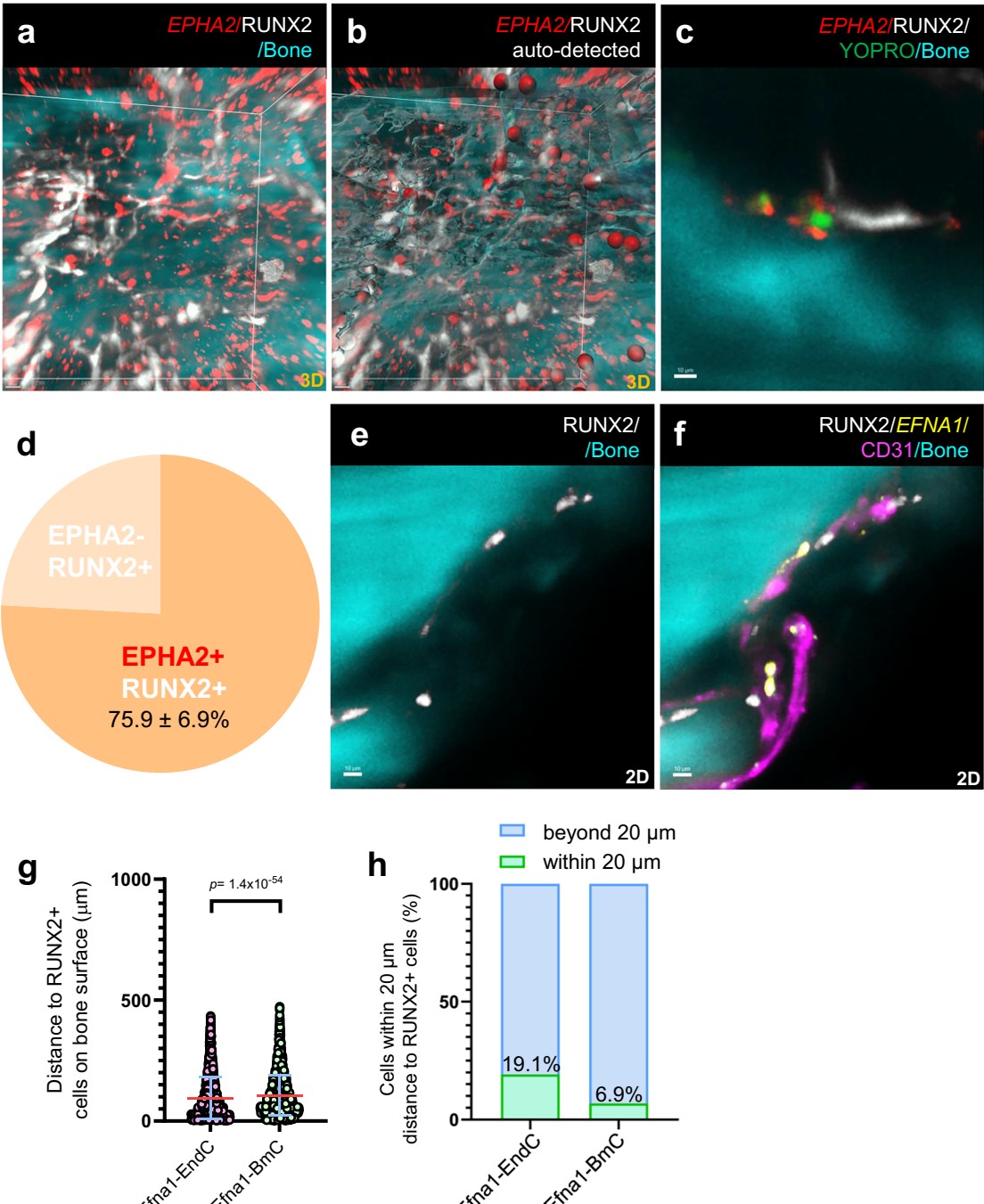

**Fig. 4 | Spatial analysis of EPHA2- and EFNA1-expressing cells. a** *EPHA2* mRNA (pseudocolored red) is shown together with RUNX2 protein (pseudocolored white) and bone matrix visualised by autofluorescence (pseudocolored cyan) in cleared human bone. A maximum intensity projection of the 3D scan is shown. **b** *EPHA2*- and RUNX2-double-positive cells are represented as red spheres, with sphere size indicating the distance to the frontal plane. Maximum intensity projection of the segmented 3D scan is shown. **c** Optical section corresponding to panel (**a**). YOPRO-stained nuclei (pseudocolored green) were omitted from panel (**a**) for visual clarity. **d** Quantification of *EPHA2*- and RUNX2-positive cells located within 20 μm of the bone surface. Data represent mean ± SE from two femoral heads (two patients), obtained from patients undergoing femoral head replacement. **e**, **f** Optical section showing RUNX2-positive cells along the bone surface (**e**), with additional channels overlaid for *EFNA1* mRNA-expressing cells (pseudocolored yellow) and CD31-positive blood vessels (pseudocolored magenta), highlighting their spatial proximity (**f**). **g** Distance from *EFNA1*-positive cells to RUNX2-positive cells on the bone surface. *EFNA1*-positive cells were stratified into two groups: those associated with blood vessels (within 20 μm, as segmented in Fig. 3f), termed Efna1-EndC (endothelial cells), and those located beyond 20 μm from both blood vessels and the bone surface, termed Efna1-BmC (bone marrow cells). *n* = 13,699 cells for Efna1-EndC and *n* = 6945 cells for Efna1-BmC, pooled from two patients within each experimental group. Data are presented as mean ± SD. Statistical analysis was performed using a two-tailed Mann-Whitney test. **h** Percentage of Efna1-EndC and Efna1-BmC cells located within 20 μm of RUNX2-positive cells detected along the bone surface. Data derived from panel (**g**). Source data are provided as a Source Data file.

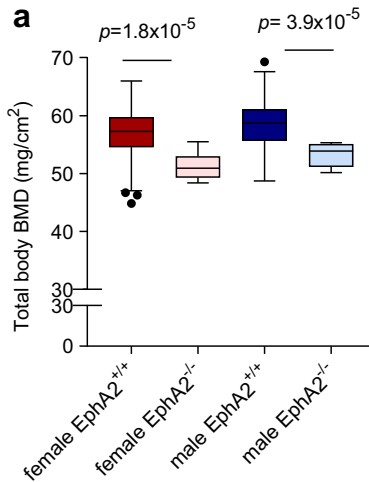
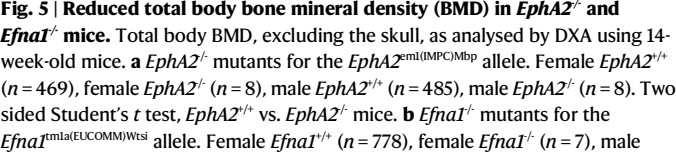
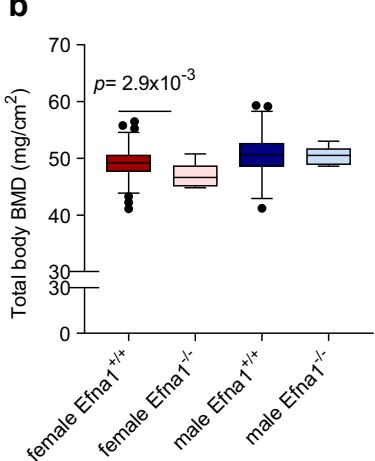

**Fig. 5 | Reduced total body bone mineral density (BMD) in *EphA2*[-/-] and *Efna1*[-/-] mice.** Total body BMD, excluding the skull, as analysed by DXA using 14-week-old mice. **a** *EphA2*[-/-] mutants for the *EphA2*[em1(IMPC)Mbp] allele. Female *EphA2*[+/+] (*n* = 469), female *EphA2*[-/-] (*n* = 8), male *EphA2*[+/+] (*n* = 485), male *EphA2*[-/-] (*n* = 8). Two-sided Student's *t* test, *EphA2*[+/+] vs. *EphA2*[-/-] mice. **b** *Efna1*[-/-] mutants for the *Efna1*[tm1a(EUCOMM)Wtsi] allele. Female *Efna1*[+/+] (*n* = 778), female *Efna1*[-/-] (*n* = 7), male

*Efna1*[+/+] (*n* = 816), male *Efna1*[-/-] (*n* = 6). Two-sided Student's *t* test, *Efna1*[+/+] vs. *Efna1*[-/-] female mice. Box and whisker plots were generated using the Tukey method for whisker calculation, where whiskers extend to 1.5 times the interquartile range (IQR) from the first and third quartiles, and outliers beyond this range are shown as individual data points. Source data are provided as a Source Data file. *DXA* dual-energy X-ray absorptiometry.

---

of DKK1-targeting treatments on BMD in humans[33]. High, genetically predicted, circulating levels of matrix extracellular phosphoglycoprotein (MEPE) were associated with reduced fracture risk, supported by colocalization evidence. This finding is in concordance with previous findings that the *MEPE* locus is associated with BMD[16] and that rare human loss of function variants in *MEPE* are associated with reduced BMD and increased fracture risk[18,19]. In the present study, high, genetically predicted, R-spondin 3 (RSPO3) was causally associated with reduced forearm fracture risk, confirming our previous results of human MR and experimental mouse studies showing that RSPO3 is crucial for bone metabolism[17].

Here, we identified three novel fracture-related proteins; CCND2, EFNA1, and TIMP2, which were causally associated with forearm fractures. The genetic signals of the three novel, fracture-related proteins were all colocalised with corresponding signals for forearm fractures, demonstrating that these proteins and forearm fractures share the same causal genetic variants. CCND2, EFNA1, and TIMP2 were causally associated with eBMD in the heel, and EFNA1 and TIMP2 were also causally associated with total body BMD, suggesting that they all exert part of their effect on fracture risk via effects on BMD.

As CCND2 is a key regulator of the G1-S transition, interfering with CCND2 function could disrupt normal cell cycle progression in multiple cell types. Therefore, we do not believe that CCND2 is a promising, specific, anti-fracture target[23,24]. TIMP2 has been proposed to modulate matrix metalloproteinases function in calvariae, where matrix metalloproteinases are known to play a role in osteoclastic bone resorption[26,27]. It was previously reported that *Timp2*[-/-] mice have reduced volumetric BMD but increased bone size[25] and a human GWAS demonstrated that a genetic signal in the *TIMP2* locus was associated with eBMD in the heel[20]. However, the present MR and colocalization findings are the first evidence of an effect of *TIMP2* on fracture risk. We observed high expression of TIMP2 in diaphyseal cortical bone, and our single-cell RNA sequencing analyses revealed high expression of *Timp2* in human osteoblast-lineage cells, supported by a previous study demonstrating that *Timp2* is expressed in mouse osteoblasts[26]. In conclusion, we provide the first evidence that high circulating levels of TIMP2 reduce BMD and increase fracture risk in humans. Our current data provide human relevance to the previously proposed role of TIMP2 in mouse bone homeostasis (Table 1). Collectively, these

findings position TIMP2 as a compelling candidate for fracture-preventing therapy.

Our MR analyses demonstrated that high circulating ephrin-A1 is causally associated with low forearm fracture risk and high BMD in humans, making ephrin-A1 a plausible novel fracture target. Therefore, we characterised the ephrin-A1 signalling pathway in bone using detailed expression analyses and functional studies. Ephrin-A1 is a ligand for the EphA receptors that represent a large family of receptor tyrosine kinases, and it is bound to the cell membrane through a glycosylphosphatidylinositol (GPI) anchor[29]. Although the most important functions of ephrin-A1 appear to be dependent on cell–cell contact using cell-membrane linked ephrin-A1, it is demonstrated that a proteolytic cleavage at the GPI region (performed by the protease ADAM12) results in a release of a soluble, monomeric form of ephrin-A1 into the circulation[34,35]. As our MR analyses demonstrated that high circulating ephrin-A1 is causally associated with low forearm fracture risk and high BMD in humans, we hypothesise that ephrin-A1, via activation of its high-affinity receptor EphA2 in bone, increases bone mass[29,30].

We used a combination of single-cell RNA sequencing, in situ hybridisation and spatial expression analyses of human bone, using the novel 3D DeepBone tissue-clearing technique, to show that ephrin-A1 is abundantly expressed in human bone marrow, including expression by CD31-positive endothelial cells in the vicinity of blood vessels. EPHA2, the high-affinity receptor of ephrin-A1[29], was also enriched in human bone and expressed in osteoblasts on the bone surface. Integrated 3D spatial expression analyses showed that EPHA2-expressing osteoblasts are closer to capillary endothelial ephrin-A1-expressing cells than to ephrin-A1-positive bone marrow cells, which suggests that direct cell-cell contact is plausible considering the short distance between these cells. An interaction between ephrin-A1 expressed by endothelial cells and its high-affinity receptor EPHA2[29] on osteoblasts was also supported by our ligand-receptor interaction analyses between clusters in the human bone marrow. The most well-established functions of ephrin-A1 are dependent on direct cell–cell interaction with cells harbouring EPHA2. Thus, these data suggest that cell-membrane-linked ephrin-A1 on CD31-positive vascular endothelial cells interacts directly with the EPHA2 receptor on osteoblasts found on the bone surfaces to increase BMD, thereby reducing fracture risk.

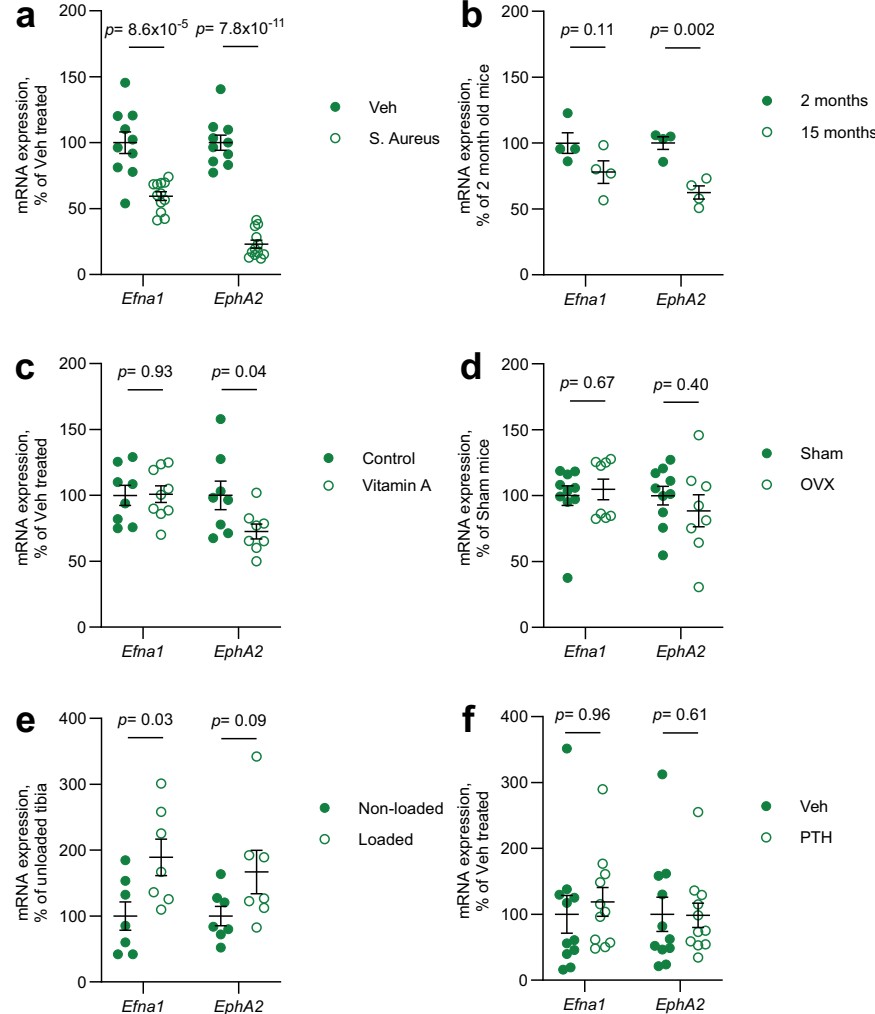

**Fig. 6 | *Efna1* and *EphA2* expression in different mouse disease models.** Relative expression of *Efna1* and *EphA2* was analysed in: (**a**) Trabecular bone from vertebrae of 10-week-old female mice systemically treated with either vehicle (Veh, *n* = 10) or *Staphylococcus aureus* (S. Aureus, *n* = 12) to induce systemic inflammation, 8 days prior to termination. **b** Trabecular bone from vertebrae of young (2-months-old, *n* = 4) and aged (15-months-old, *n* = 4) female mice. **c** Cortical bone from tibiae of 9-week-old female mice fed either a control chow diet (Control, *n* = 8) or chow supplemented with high-dose vitamin A (*Efna1*, *n* = 9; *EphA2*, *n* = 8) for 4 days. **d** Cortical bone from tibiae of 17-week-old female mice subjected to sham surgery (Sham, *n* = 10) or ovariectomy (OVX, *n* = 8) at 13 weeks of age. **e** Cortical bone from

tibiae of 15-week-old female mice subjected to mechanical loading of the right tibia (*n* = 7), with the contralateral (left) tibia serving as non-loaded control (*n* = 7). **f** Cortical bone from tibiae of 14-week-old male mice treated with vehicle (Veh, *n* = 11) or parathyroid hormone 1–34 (PTH, *n* = 11), administered intraperitoneally 5 days per week for 3 weeks prior to termination. Data are presented as percentage relative to respective control groups, with individual values shown as scatter dot plots. Lines indicate mean ± standard error. Statistical analysis was performed using a two-sided Student's *t* test. Source data are provided as a Source Data file. *Veh* vehicle; *OVX* ovariectomy; *PTH* parathyroid hormone.

A similar direct cell-cell interaction between vascular endothelial cells and osteoblast-lineage cells is well-established for Notch signalling, which is crucial for bone homeostasis[36,37]. The direct contact between endothelial cells of blood vessels and osteolineage cells is likely facilitated by the presence of sinusoidal capillaries. These capillaries have very thin basement membranes that are often absent, allowing blood cells to migrate in and out of the bloodstream. We recently showed that sinusoidal capillaries are lining the bone surface in humans[28], further ensuring this possibility. A functional role for ephrin-A1–EphA2 signalling in bone homeostasis is supported by our functional observations that both *Efna1*−/− and *EphA2*−/− mice had reduced BMD.

In the MR analyses, we identified high levels of circulating ephrin-A1 as causally linked to reduced fracture risk and increased BMD. We believe that the circulating levels of ephrin-A1 may reflect the ephrin-A1 expression by endothelial cells, which are abundant throughout the body. Whether endothelial cells express EFNA1 ubiquitously or

selectively in specific organs, like in the liver and bone, observed in the present study, remains to be determined. However, since bone is an abundantly vascularises, hematopoietic organ it appears plausible that the observed high expression of ephrin-A1 by vascular endothelial cells close to osteoblasts in bone, contributes to a local beneficial effect of ephrin-A1 on bone homeostasis. This is analogous to the present and previous MR findings demonstrating that high circulating sclerostin levels are causally associated with increased fracture risk and reduced BMD, despite the well-established role of osteocyte-derived sclerostin in the local regulation of bone homeostasis[38]. However, we cannot exclude a contributing role of circulating or paracrine ephrin-A1 for bone homeostasis.

To further elucidate the cellular origin of EFNA1 relevant to bone health, future functional studies employing novel mouse models with conditional inactivation of EFNA1 in endothelial cells and other candidate cells, as well as models with endothelial-specific overexpression of EFNA1, will be required.

To the best of our knowledge, no prior studies have investigated ephrin-A1–EphA2 signalling as a potential therapeutic target for fracture prevention. However, previous in vitro studies have indicated a bidirectional interaction between osteoclastic ephrin-A2 and osteoblastic EphA2, promoting osteoclastogenesis while concurrently suppressing osteoblastic bone formation[39]. The role of ephrin-B2–EphB4 signalling in bone homeostasis is more firmly established[39]. Experimental evidence has shown that the interaction between osteoclastic ephrin-B2 and osteoblastic EphB4 suppresses bone resorption while promoting osteoblastic bone formation[39]. Furthermore, in vivo studies by Bikle et al. have demonstrated that ephrin-B2–EphB4 signalling mediates the effects of insulin-like growth factor I (IGF-I) in the regulation of endochondral bone formation[40]. The same research group also demonstrated that ephrin-B2 expression in Col2-positive cells plays a critical role in regulating fracture repair[41]. Thus, although other ephrins have been implicated in bone homeostasis and fracture repair in experimental settings, the present human MR-colocalization pipeline identifies ephrin-A1 as a key regulator of fracture risk in humans. Future studies are warranted to elucidate the potential role of ephrin-A1–EphA2 signalling in the regulation of fracture repair.

The data presented in this study suggest that membrane-bound ephrin-A1 expressed on vascular endothelial cells may directly interact with the EphA2 receptor on osteoblasts. We hypothesise that this ephrin-A1–EphA2 signalling promotes osteoblast-mediated bone formation, thereby contributing to increased bone mineral density. The drug targets underlying currently approved osteoporosis treatments were primarily identified as mechanisms with an effect on BMD[12]. Importantly, the heritable component of fracture risk is partly independent of BMD[42,43]. It is therefore plausible that key druggable targets for fracture prevention have remained undiscovered due to prior focus on mechanisms affecting BMD. This notion is further supported by our recent large-scale GWAS on forearm fractures, which identified several genetic loci associated with bone quality parameters that were not detected in previous GWAS focused on BMD[12]. Given that the current MR-colocalization pipeline utilises the same forearm fracture GWAS for outcome analyses[12], we hypothesise that this approach enables the identification of both BMD-dependent and BMD-independent mechanisms contributing to fracture risk. Although we propose that a portion of the effect of ephrin-A1 on fracture risk may be mediated through alterations in BMD, the underlying mechanism for ephrin-A1 on fracture risk may also involve yet unidentified influences on bone quality.

Our comprehensive descriptive analyses revealed a marked reduction in the expression of ephrin-A1 and/or its high-affinity receptor EphA2 in bone tissue in murine models of bone loss induced by inflammation, aging, or high-dose vitamin A administration. In contrast, their expression remained unchanged in a model of bone loss induced by ovariectomy. Moreover, the anabolic bone response elicited by mechanical loading, but not by intermittent parathyroid hormone (PTH) administration, was associated with upregulated expression of ephrin-A1 in bone tissue. The observed pattern of altered ephrin-A1 and EphA2 expression in selected osteoporosis disease models, as well as in response to certain anabolic stimuli, suggests that ephrin-A1–EphA2 signalling in bone may represent a mechanistically distinct pathway from those targeted by currently approved osteoporosis therapies. Based on these findings, we hypothesise that therapeutic strategies targeting ephrin-A1–EphA2 signalling may act synergistically with existing antifracture treatments, offering complementary and potentially additive benefits in the management of osteoporosis. Furthermore, therapeutic modulation of ephrin-A1–EphA2 signalling may not be associated with the increased risk of serious, albeit rare, adverse events linked to long-term use of currently approved osteoporosis medications[4,5]. This could potentially enhance patient adherence to treatment.

One limitation of the present study is that, despite the exclusive use of cis-pQTLs as genetic instruments, establishing their validity with complete certainty remains challenging.

In conclusion, ephrin-A1–EphA2 signalling is a novel potential target for fracture prevention. We propose that ephrin-A1 on endothelial cells interacts directly with EphA2 on osteoblasts to increase BMD, thereby reducing fracture risk.

## Methods

### Mendelian randomisation
To identify proteins that are causally associated with forearm fractures, we used previously published GWASs for circulating proteins[7] and forearm fractures[12]. We selected non protein-altering cis-pQTLs detected in the UK Biobank using the Olink proteomics platform[7]. We restricted the selection to only include SNPs with a minor allele frequency (MAF) above 1% that also were available in the forearm fracture GWAS. To ensure that we could harmonise all SNPs we removed palindromic SNPs with MAF between 0.45-0.55. To ensure that we only included independent SNPs, we performed LD-pruning with an $r^2$ threshold of 0.01, keeping the most significant pQTLs for each protein. The SNPs were harmonised to have the same effect allele in both protein and outcome data. MR analyses were performed for each protein using the mr_ivw function in the MendelianRandomization R-package. This function uses the Wald ratio when there is only one SNP available and inverse variance weighted MR when there are more than one SNP available, with fixed or random effects depending on the results from the Cochran's Q test of heterogeneity.

### Colocalization
Colocalization analyses between forearm fracture and the nine selected proteins were performed using PWCoCo (https://github.com/jwr-git/pwcoco) with reference data from 10,000 randomly selected unrelated participants from the UK Biobank. PWCoCo implements conditional analyses from GCTA-COJO[44,45] and colocalization analyses from the R coloc-package[46] to allow for unbiased colocalization evidence. The initial analysis is performed on unconditional summary statistics for both traits. When there is evidence of colocalization on unconditional summary statistics (PP.H4 ≥ 0.8), no conditional analyses are performed. When PP.H4 < 0.8, PWCoCo performs conditional analyses of the summary statistics for both traits. If there is only one conditionally independent SNP in the region, this SNP will be conditioned upon. If there are multiple conditionally independent SNPs PWCoCo will isolate one SNP at the time by conditioning upon the others. Colocalization analyses are then performed on all combinations of unconditional and conditional summary statistics for the two traits.

### Evaluation of total body BMD in *Efna1*$^{-/-}$ and *EphA2*$^{-/-}$ mice
*Efna1*$^{-/-}$ mice (*Efna1*$^{tm1a(EUCOMM)Wtsi}$ allele) and *EphA2*$^{-/-}$ mice(*EphA2*$^{em1(IMPC)Mbp}$ allele) were produced by the International Mouse Phenotyping Consortium (IMPC) (www.mousephenotype.org) and analysed at 14 weeks of age[47]. Total body BMD (excluding skull) for the *EphA2*$^{-/-}$ mutants was measured by the UC Davis phenotyping centre using a DEXA (Dual Energy X-ray Absorptiometry) analyser (UltraFocus DXA, Faxitron Bioptics LLC, US). Total body BMD for the *Efna1*$^{-/-}$ mutants was measured by the WTSI centre using a Lunar PIXImus 2 DEXA analyser (GE LUNAR, US). Publicly available data of total body BMD for the *EphA2*$^{-/-}$, *Efna1*$^{-/-}$, and control mice were downloaded from IMPC January 31, 2025.

### Animal experiments at University of Gothenburg
All animal procedures performed at the University of Gothenburg were approved by the Gothenburg Animal Research Ethics Committee, Sweden, and the animals were cared for according to their guidelines. Sex was not specifically considered in the study design or analysis;

however, all data are reported disaggregated by sex. The animals were housed in an animal house at the University of Gothenburg, under a controlled temperature (20 °C), and photo period (12 h of darkness and 12 h of light). Pellet diet (Teklad diet 2016, Envigo, Indianapolis, IN, US), unless otherwise stated, and water was available ad libitum. At the end of each experiment, mice were euthanised using Ketador (Richter Pharma) and Dexdomitor (Orion Pharma), followed by exsanguination and cervical dislocation. When applicable, soft tissues were dissected and snap-frozen in liquid nitrogen, whereas cortical bone (from clean and flushed tibial shafts) and trabecular bone (from vertebral bodies) were placed in RNAprotect Tissue Reagent (76106, Qiagen) and stored at − 80 °C until RNA extraction, as described below.

### Tissue-specific expression profiling

In order to determine the expression of *Efna1* and *EphA2* in different tissues, twelve-week-old female and male C57BL/6 N wildtype mice were euthanised and tissues were collected.

### Mouse models for bone loss and anabolic bone formation

To investigate whether *Efna1* and/or *EphA2* expression in bone tissue is regulated in osteoporosis disease models or by bone anabolic treatments, a variety of mouse models were used:

**A systemic inflammation-induced bone loss model.** To induce systemic inflammation, 9-week-old female C57BL/6 N mice were intravenously inoculated via the tail vein with 0.2 ml of Staphylococcus aureus (SA, $5 \times 10^7$ CFU/mouse) or PBS as vehicle control (Veh). Eight days post-inoculation, the mice were euthanised, and the vertebrae were dissected. Microcomputer tomography analysis revealed a significant reduction in bone volume over total volume (BV/TV) in the L5 vertebra of SA-treated mice ($-29.5 \pm 2.4\%$ compared to vehicle controls; Student's *t* test, $p = 1.9 \times 10^{-7}$).

**An age-induced bone loss model.** 15-month-old and 2-month-old female mice from the same strain (C57BL/6JRj) and breeder (Janvier Labs, France) were euthanised, and the vertebrae were dissected. A recent publication from our group demonstrates a significant reduction of trabecular bone volume fraction following aging[48].

**A vitamin A-induced bone loss model.** Nine-week-old female C57BL/6 N mice were fed either a control chow diet (Control, Teklad diet 2016, Teklad Custom Diets, Madison, WI) or a chow supplemented with a high dose of retinyl acetate (vitamin A, 700 IU/g; Teklad Custom Diets). After four days of dietary intervention, the mice were euthanised, and tibia and femur were dissected. Peripheral quantitative computed tomography analysis of the femur revealed a significant reduction in the cortical thickness following high-dose vitamin A supplementation ($-6.0 \pm 1.7\%$ compared to control fed mice; Student's *t* test, $p = 1.7 \times 10^{-2}$).

**A postmenopausal osteoporosis model.** 13-week-old C57BL/6 N female mice underwent ovariectomy or sham surgery. Four weeks post-surgery, the mice were euthanised, and tibia and femur were dissected. Peripheral quantitative computed tomography analysis of the femur revealed a significant reduction in the cortical thickness following ovariectomy ($-7.0 \pm 1.5\%$ compared to sham mice; Student's *t* test, $p = 3.7 \times 10^{-4}$). Some data from these mice have been published previously (unrelated to the present study)[49].

**A loading-induced bone formation model.** We employed a loading-induced bone formation model using 13-week-old female C57BL/6 N mice fed standard chow (TD.00217, Envigo, Huntingdon, UK). For 3 days a week for 2 weeks, the right tibia was subjected to loading by placing the tibia in the holding cups of the ElectroForce 3100 Test Instrument, under the sedation of isoflurane (Baxter). Axial load of

14.4 N was applied through the knee joint for 40 cycles with 10 s rest between the cycles. The left unloaded tibia served as control. As previously demonstrated, loading increased cortical thickness ($+25.2 \pm 1.6\%$ compared to the non-loaded bone; Student's t test, $p = 9.3 \times 10^{-10}$)[50].

**A PTH-induced bone formation model.** 11-week-old male mice were treated with either vehicle (Veh, 20 mM $NaH_2PO_4$ in NaCl) or human parathyroid hormone 1-34 (PTH, dissolved in 20 mM $NaH_2PO_4$ in NaCl, Bachem). Treatments were administered via intraperitoneal injection 5 days a week for 3 weeks. Following the treatment period, mice were euthanised, and tibiae were dissected. Peripheral quantitative computed tomography analysis of the tibia revealed a significant increase in cortical thickness in PTH-treated mice ($+11.7 \pm 1.6\%$ compared to vehicle-treated control mice; Student's *t* test, $p = 6.2 \times 10^{-3}$).

### Real-time quantitative PCR

Total RNA from trabecular-rich (vertebral body) bone, cortical (shafts from flushed tibia), muscle soleus, retroperitoneal fat, gonadal fat, brown fat, muscle gastrocnemius, bone marrow, cerebral cortex, hypothalamus, levator ani, and pituitary gland was prepared using TRIzol Reagent (15596018, Thermo Fisher Scientific), followed by the RNeasy Mini Kit (74116, Qiagen). Total RNA from liver, kidney, lung, jejunum, heart, ovary, testis, prostate, vesicle seminalis, aorta, uterus, adrenal gland, spleen, proximal colon, and thymus was prepared using the RNeasy Mini Kit. To obtain cDNA, the RNA was reversed transcribed (4368814, Applied Biosystems) and real-time PCR analyses were performed using the QuantStudio 3 Real-Time PCR System (Thermo Fisher Scientific) using the following Assay-on-Demand primer and probe set: *Efna1* (encoding *Ephrin A1*), Mm01212795_m1; *EphA2* (encoding Eph receptor A2), Mm00438726_m1; and *Timp2* (encoding Tissue inhibitor of metalloproteinases 2), Mm00441825_m1. Relative gene expression was calculated using the $2^{-\Delta\Delta Ct}$ method using the ribosomal subunit 18S as an internal standard.

### Single-cell RNA sequencing data processing

To evaluate gene expression in different rare human non-hematopoietic bone marrow populations, a special atlas was created by combining several single-cell RNA sequencing experiments enriched for cell types of interest[28]. For the present study, raw expression matrices from separate experiments were concatenated (https://www.ncbi.nlm.nih.gov/geo/query/acc.cgi?acc=GSE147287; https://www.ncbi.nlm.nih.gov/geo/query/acc.cgi?acc=GSE147390; https://www.ncbi.nlm.nih.gov/geo/query/acc.cgi?acc=GSE169396; https://www.ncbi.nlm.nih.gov/geo/query/acc.cgi?acc=GSE190965; https://www.ncbi.nlm.nih.gov/geo/query/acc.cgi?acc=GSE196678; https://www.ncbi.nlm.nih.gov/geo/query/acc.cgi?acc=GSE202813)[51–54]. After quality control, data transformation, and highly variable gene selection, PCA-based dimensionality reduction was applied. To adjust for batch effects at the level of PCs, we leveraged the Harmony approach[55]. Adjusted PCs were utilised to construct a neighbour graph, which connects cells with similar transcriptomes. Leiden clustering and UMAP embedding of the data were based on this graph. Cell types were annotated by assessing expression levels of known RNA markers (Supplementary Fig. 7a). For gene expression embeddings, the number of transcripts per 10 K total transcripts per cell are displayed, with values capped at the 99.8 quantile to prevent outliers from distorting contrast levels. We relied on the Python-based Scanpy and Anndata (python 3.9, anndata 0.9.0, liana 1.1.0, matplotlib 3.8.4, numpy 1.26.4, pandas 2.3.3, scanpy 1.10.1, pymultimap 0.0.8, scvi 0.6.8), toolkits for the analysis[56].

### Single-cell RNA sequencing ligand-receptor interaction analysis

To study putative ligand-receptor interactions, we used the LIANA package[32], which aggregates outputs from a set of established and

widely used methods and returns consensus results, reflecting relative specificity and strength (magnitude) of known ligand-receptor pairs expression between different cell types. The analysis was restricted to endothelial, stromal, and osteolineage cells. From the outputs, we selected interactions with EFNA1 as the ligand and EPHA receptors as the target, retaining those with *P*-values < 0.05.

### In situ hybridisation

Multiplex fluorescence in situ hybridisation (FISH) and chromogenic in situ hybridisation (CISH) of mouse bone tissue were performed using the RNAscope Multiplex Fluorescent V2 Assay (Advanced Cell Diagnostics (ACD); 323100) and the RNAscope Duplex Detection Kit (ACD; 322500), respectively, according to the manufacturer's protocols (ACD). The following probes were used: Mm-Efna1-C1 (ACD; 428621), Mm-EphA2-C2 (ACD; 313221-C2), Mm-Runx2-C2 (ACD; 414021-C2), Mm-Pecam1-C3 (ACD; 316721-C3; coding for CD31), Mm-Cxcl12-C3 (ACD; 422711-C3), and Mm-Acp5 (ACD; 465001-C3). For FISH, signal amplification was achieved using TSA Plus fluorophore kits (PerkinElmer; NEL741001KT, NEL744001KT, NEL745001KT) at a 1:800 dilution, followed by DAPI counterstaining. For CISH, signals were detected using fast red (C2 channel) and green (C1 channel) chromogens included in the kit, and sections were counterstained with 50% haematoxylin. Imaging was conducted using a Nikon spinning disk confocal microscope for FISH and a Zeiss slice scanner microscope for CISH.

### 3D DeepBone labelling of *EFNA1, EPHA2, CD31, RUNX2,* and *CD56*

**Pretreatment of samples.** Samples were obtained from patients undergoing femoral head replacement (two women with hip fractures and one man with osteoarthritis). After surgical removal of the femoral head, needle biopsies (3-4mm in diameter) were obtained from the patient's femoral head 0–2cm beneath the cartilage. The biopsies were fixed in 4% paraformaldehyde overnight at 4 °C with gentle agitation. The needle biopsies were washed with 2X SSCT (saline-sodium citrate/tween-20) for 1 h at 4 °C, followed by dehydration to 60% methanol (MeOH) sequentially with 20%, and 40% MeOH for 1 h each at 4 °C with gentle agitation. Dehydrated biopsies were decolorised with 3% $H_2O_2$/20% DMSO/70% MeOH overnight at 4 °C. The next day, the samples were dehydrated to 100% MeOH for 1 hour and delipidated with dichloromethane (DCM) for 1 hour and overnight at 4 °C. Next day, the samples were washed with DCM again for 1 hour for a total of at least 3 washes. The biopsies were then re-hydrated with 100%, 80%, 40% MeOH for 1 hour. Followed by washing in 2X SSCT buffer, the bone biopsy was decalcified in 20% formic acid 10% sodium citrate overnight and subsequently washed twice with 2X SSCT buffer, 1 h each, and proceed to RNA and protein labelling.

**3D RNA and protein labelling of bone biopsies.** After decolorisation, delipidation and decalcification, *EFNA1* and *EPHA2* were detected by Hybridisation Chain Reaction (HCR) following the manufacturer's instructions. Briefly, the samples were labelled with oligo arrays of *EFNA1* and *EPHA2* probe pairs synthesised by IDT (Supplementary Data 6). The hybridised oligos were washed and amplified by fluorescent conjugated oligo amplifiers, which form a polymer on the hybridised mRNA, and labelled cells were detected using lightsheet microscopy. Details of the protocol has previously been reported[57].

**3D protein labelling of vasculature.** After RNA labelling, the SSCT were washed twice with 1X PBST/25 mM EDTA, for 1 h each, to normalise the salt concentration for antibody labelling. The samples were blocked for 4–6 h at room temperature using blocking buffer (5% donkey serum, 10% DMSO, 0.2% TritonX-100, 25 mM EDTA in 1X PBS). After blocking, the samples were labelled with the antibodies at 37 °C for two days, followed by four washes of antibodies with 1XPBST/25 mM EDTA. The following primary antibodies were used: anti-CD31 (AF3628, R&D systems, 1:100 dilution), anti-RUNX2 (ab192256, Abcam,

1:100), and anti-CD56 (AF2408, R&D systems, 1:100 dilution). The corresponding secondary antibodies were applied thereafter for two days.

**Lightsheet microscopy, image acquisition and image analysis.** Lightsheet microscopy was performed using Ultramicroscope Blaze (Miltenyi Biotec). Images were taken with 4 μm interval to capture individual cells without losing information (assuming average cell size ~10 μm in diameter). Autofluorescence was acquired with 405 nm excitation laser. Cellular resolution was achieved with 3.2X with a 0.944 pixel/micron ratio. Raw scans were acquired as TIFF format and were compiled into 3D stacks by ImarisConverter 10.0. The Imaris were then processed with Imaris 10.0. During image processing, individual cells were first identified by a volume filter excluding signals that are too large ($> 20,000 \, \mu m^3$) or too small ($< 500 \, \mu m^3$). Individual cells were identified by spot function with a spot function with an estimate of 12 μm diameter. CD31-positive vasculature was identified by surface function using CD31 signals excluding spherical surface (sphericity < 0.7). The bone surface was identified by a supervised machine learning algorithm using the surface function available in Imaris 10.2[58,59]. For the integrated spatial analysis, cells were stratified based on their distance from the corresponding surface, using a 20 μm threshold. Specifically, EPHA2-positive cells located within 20 μm of the bone surface mask were classified as "EPHA2-positive cells on the bone surface", while EFNA1-positive cells within 20 μm of the vessel surface mask were considered vessel-associated. All spot-to-surface and spot-to-spot distances were calculated using the "shortest distance between spots" function in Imaris 10.2, with measurements consistently taken from the centre of each spot.

### Ethics oversight

For the human Mendelian randomisation analyses, we used publicly available GWAS summary statistics. These GWASs have previously been published with relevant ethical approvals[7,12,20,22]. The mechanistic mouse studies were approved by the regional animal ethics committee in Gothenburg. The histology work with human bone samples was approved by the Swedish Ethics Review Authority, and written informed consent was obtained from all patients (Etikprövningsmyndigheten Dnr 2022-01977-02).

### Statistics & reproducibility

Statistical analysis of mice data was performed using GraphPad Prism v10.4.0. No statistical method was used to predetermine sample size. All data were included in the analyses unless samples were excluded due to technical issues, the number of replicates is provided in each figure or table. Experiments were not randomised, and investigators were not blinded to allocation during experiments and outcome assessment.

### Reporting summary

Further information on research design is available in the Nature Portfolio Reporting Summary linked to this article.

## Data availability

All GWAS summary statistics for the exposures and outcomes in the Mendelian randomisation analyses are available online: circulating proteins https://www.decode.com/summarydata/, eBMD and total body BMD http://www.gefos.org/, forearm fractures at the GWAS Catalogue under study accession number GCST90281273 (https://www.ebi.ac.uk/gwas). The human bone marrow single-cell RNA sequencing atlas can be explored at the CellxGene Portal:https://cellxgene.cziscience.com/collections/0391c84c-d57d-4741-9277-e4d58f9a3d0c. and primary data used in the present study are available at https://www.ncbi.nlm.nih.gov/geo/query/acc.cgi?acc=GSE147287; https://www.ncbi.nlm.nih.gov/geo/query/acc.cgi?acc=GSE147390; https://www.ncbi.nlm.nih.gov/geo/query/acc.cgi?acc=

GSE169396; https://www.ncbi.nlm.nih.gov/geo/query/acc.cgi?acc= GSE190965; https://www.ncbi.nlm.nih.gov/geo/query/acc.cgi?acc= GSE196678; https://www.ncbi.nlm.nih.gov/geo/query/acc.cgi?acc= GSE202813. Source data are provided in this paper.

## Code availability

All analyses were performed using publicly available software, tools, packages and databases. The MR analyses were conducted using R v4.4.3 (https://cran.r-project.org/) and the packages MendelianRandomization, LDlinkR, and dplyr. Colocalization analyses were performed using the pwcoco tool (https://github.com/jwr-git/pwcoco).

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

## Acknowledgements

This work was supported by the Swedish Research Council (2020-01392 and 2024-02412, C.O.; 2022-01156, S.M.-S.; 2020-02298, A.S.C.); the Swedish state under the agreement between the Swedish government and the county councils, the ALF-agreement (ALFGBG-720331, ALFGBG-965235, and ALFGBG-965744, C.O.; ALFGBG-1005555, S.M.-S.; ALFGBG-1006264, A.S.C.); the Novo Nordisk Foundation (NNF19OC0055250 and NNF22OC0078421, C.O.; NNF23OC0084522 and NNF25OC0105187, S.M.-S.; NNF21OC0070314, A.S.C.); the Lundberg Foundation (LU2024-0110, C.O.; LU2025-0079, A.S.C.); the Knut and Alice Wallenberg Foundation (KAW 2020.0230, C.O.); and the European Union (ERC Advanced Grant, HeMaFA, Project 101096347, C.O.). Views and opinions expressed are, however, those of the author(s) only and do not necessarily reflect those of the European Union or the European Research Council. Neither the European Union nor the granting authority can be held responsible for them. The funders had no role in study design, data collection, data analysis, data interpretation, or manuscript writing.

## Author contributions

The design of the study was performed in collaboration between S.M.-S., M.N. and C.O. Mendelian randomisation and colocalization were conducted by M.N. and C.O. Mechanistic studies were conducted by S.M.-S., K.H.N., P.H., U.H.L., and C.O. L.L. conducted the in situ hybridisation using mouse tissues, O.D. and A.S.C. conducted the human bone marrow single-cell RNA seq analyses, also including ligand-receptor interaction analyses. N.T.L.C., X.T. and A.S.C. conducted the analyses of human bone using 3D spatial expression. S.M.-S., M.N., A.S.C. and C.O. wrote the first draft of the manuscript. All authors contributed to subsequent drafts of the manuscript and made the decision to submit the manuscript for publication.

## Funding

## Competing interests
C.O. is an applicant on filed patent applications on the effect of probiotics on bone metabolism. All other authors declare no competing interests.
