## [Transparent Peer Review file · Nature Communications]

Identification of ephrin-A1–EphA2 signalling as a potential target for fracture prevention

Corresponding Author: Professor Claes Ohlsson

Version 0:

Reviewer comments:

Reviewer #1

(Remarks to the Author)

The authors performed a proteome-wide Mendelian randomization (MR) and colocalization study to investigate potential targets for fracture prevention. Their subsequent mechanistic work, which combined single-cell RNA sequencing, in situ hybridization, and spatial expression analyses of human bone, revealed abundant expression of ephrin-A1 in bone marrow. Based on these findings, they propose ephrin-A1-EphA2 signaling as a novel therapeutic target for preventing fractures. Several aspects of the study need further consideration.

1. While the authors claim their MR analysis used valid genetic instruments, this statement may not be precise enough. First, the identification of FAM3C as a false positive raises concern about potential pleiotropy or confounding factors that could challenge the IV validity. Second, the validity of instrumental variable assumptions remains difficult to empirically verify in this study design.
2. The conclusion that bone mineral density (BMD) mediates the effects of identified proteins (CCDN2, TIMP2, EFNA1) on fracture risk appears premature. While the MR analysis shows an association between these proteins and BMD, this alone does not establish causal mediation.
3. To strengthen their findings, the authors should consider replicating their MR analyses using independent protein quantitative trait loci datasets and fracture/BMD GWAS data from different cohorts or conduct sensitive analysis. Demonstrating consistency across diverse cohorts and through alternative analytical methods would significantly enhance confidence in the reported causal relationships.
4. To better contextualize these findings within current research, could the authors provide more detailed insights into recent developments regarding ephrin-A1-EphA2 signaling as a target for fracture prevention?
5. A minor issue: inconsistent notation: "EphA2" and "Epha2" are used in the manuscript.

Reviewer #2

(Remarks to the Author)

What are the noteworthy results?

Based on a large genetic patient screening of osteoporotic forearm fracture cases the author group presents several known genes that are currently clinically targeted by approved drugs (sclerostin and osteoprotegerin) and three other proteins (DKK1, RSPO3, and MEPE) are well established bone-related proteins and a group of three so far unknown genes that were now analyzed in more detail. In subsequent mechanistic studies, they focused on one of these novel proteins, ephrin-A1, they found to be specifically associated with forearm fractures.

Will the work be of significance to the field and related fields?

Conceptually, the work presents on the one hand an interesting approach to identify so far unknown novel targets. These targets may indeed be interesting to follow and understand their role in more depth and compare to the established understanding of osteoporosis. On the other hand, the work fails to bring the novel target in a relationship to bone

homeostasis on a more general level or on the specific level of known derailed homeostasis in osteoporosis. The link between pre-osteoblasts and endothelial cells and their direct communication on the bone marrow niche and on the endothelial side is not novel and has been earlier described by Adams et al and others. How would this interaction be derailed in osteoporotic patient settings and how would such derailed setting impact bone mineral density or bone mass? Would it directly impact osteoblasts bone formation capacity or the indirect link through an increased osteoclasts activity? Also, would this novel path be competing or complementing the (known) osteoporotic derailments that current treatments aim to address.

Does the work support the conclusions and claims, or is additional evidence needed?

I am struggling with the claim of prevention that the ephrin-A1 should have. Prevention is very hard to show and prove. I would suggest putting this novel target (ephrin-A1) in relation to the known and druggable targets... I would strongly suggest to re-consider the current manuscript punch line and relate the work to established osteoporotic strategies and concepts and thereby open the possibility of a novel perspective.

Reviewer #3

(Remarks to the Author)

This article presents a compelling investigation of circulating proteins associated with fracture risk using MR and colocalization analysis. A significant highlight is the novel 3D DeepBone tissue-clearing technique which allows for detailed spatial analysis of EFNA1 and EPHA2 localization in bone tissue. The findings regarding EFNA1's interaction with EPHA2 provide valuable insights into the mechanisms underlying bone fractures and their prevention.

Suggestions for Improvement

1. While the authors suggest that EFNA1 may primarily be expressed by vascular endothelial cells, there is a lack of direct experimental evidence to support this claim. I recommend adding experiments that include overexpressing EFNA1 in endothelial cells and assessing its effects on osteoblast lineage differentiation and mineralization. Additionally, performing conditional knockout of the EFNA1 gene in vascular endothelial cells to observe the impact on bone mass would be beneficial.
2. Including a longitudinal assessment of EFNA1 and EPHA2 levels in various fracture risk models would be advantageous. This approach could help clarify whether elevated levels of these proteins serve as biomarkers for increased fracture risk over time.
3. The authors should also consider discussing the clinical implications of their findings, such as potential therapeutic targets for fracture prevention. Exploring how modifying EFNA1 or EPHA2 expression might influence fracture healing or risk in clinical settings would enhance the relevance of this research.
4. Overall, this is an excellent study that opens new avenues for research on EFNA1 and its role in bone health.

Version 1:

Reviewer comments:

Reviewer #1

(Remarks to the Author)

Thank the authors for their efforts to address my comments.

Reviewer #2

(Remarks to the Author)

thank you for addressing my concerns - I have no further comments!

Reviewer #3

(Remarks to the Author)

I have re-reviewed the revised manuscript. The authors have thoroughly addressed the reviewers' comments and made substantive improvements to the manuscript. The detailed assessment of ephrin-A1/EphA2 expression across multiple bone-loss and bone-anabolic models (new Figure 6) is particularly valuable, and the authors have clearly outlined necessary future functional studies in the Discussion. While additional cell-type-specific functional validation would be desirable, the current evidence provides sufficient merit for publication. I recommend acceptance of the manuscript.

REVIEWER COMMENTS

Reviewer #1 (Remarks to the Author):

The authors performed a proteome-wide Mendelian randomization (MR) and colocalization study to investigate potential targets for fracture prevention. Their subsequent mechanistic work, which combined single-cell RNA sequencing, in situ hybridization, and spatial expression analyses of human bone, revealed abundant expression of ephrin-A1 in bone marrow. Based on these findings, they propose ephrin-A1-EphA2 signaling as a novel therapeutic target for preventing fractures. Several aspects of the study need further consideration.

1. While the authors claim their MR analysis used valid genetic instruments, this statement may not be precise enough. First, the identification of FAM3C as a false positive raises concern about potential pleiotropy or confounding factors that could challenge the IV validity. Second, the validity of instrumental variable assumptions remains difficult to empirically verify in this study design.

AU: To minimize the risk of false-positive findings, we developed an integrated analytical pipeline that incorporates both MR evidence and colocalization analysis (Fig 1A). Although the FAM3C signal was initially identified in the MR step, it did not meet the criteria for colocalization and was therefore classified as a false-positive by our combined pipeline. Consequently, we did not consider FAM3C a novel candidate fracture-associated protein. In contrast, three other signals—CCND2, TIMP2, and EFNA1—demonstrated both robust MR associations and supportive colocalization evidence.

Although we exclusively used cis-pQTLs as genetic instruments, we acknowledge that it remains challenging to fully validate their suitability. We have now included this as a limitation of the present study.

Discussion (Page 16, Line 414)

“One limitation of the present study is that, despite the exclusive use of cis-pQTLs as genetic instruments, establishing their validity with complete certainty remains challenging.”

2. The conclusion that bone mineral density (BMD) mediates the effects of identified proteins (CCDN2, TIMP2, EFNA1) on fracture risk appears premature. While the MR analysis shows an association between these proteins and BMD, this alone does not establish causal mediation.

AU: We agree, and we have now tuned down this statement.

Abstract

Old version: “Both *Efna1*^{-/-} and *EphA2*^{-/-} mice had reduced bone mineral density (BMD) and explorative MR showed that ephrin-A1 increases BMD in humans, suggesting that part of the effect of ephrin-A1 to reduce fracture risk is mediated via BMD”.

New version: “Both *Efna1*^{-/-} and *EphA2*^{-/-} mice had reduced bone mineral density (BMD) and explorative MR showed that ephrin-A1 increases BMD in humans, suggesting that part of the effect of ephrin-A1 to reduce fracture risk may be mediated via BMD”.

Results (Page 6, Line 118)

Old version: “These findings suggest that these three proteins exert at least part of their effects on fracture risk via effects on BMD.”

New version: “These findings suggest that these three proteins may exert some of their effects on fracture risk via effects on BMD.”

3. To strengthen their findings, the authors should consider replicating their MR analyses using independent protein quantitative trait loci datasets and fracture/BMD GWAS data from different cohorts or conduct sensitive analysis. Demonstrating consistency across diverse cohorts and through alternative analytical methods would significantly enhance confidence in the reported causal relationships.

AU: We conducted additional MR sensitivity analyses using alternative genetic instruments for EFNA1. First, we used a different protein quantitative trait locus (pQTL) identified in the same protein GWAS based on the Olink proteomic platform.¹ This variant was excluded from the primary analyses due to linkage disequilibrium ($R^2 > 0.01$) with the lead genetic instrument for EFNA1. Second, we utilized another genetic instrument derived from a separate large-scale proteome-wide GWAS², which also included EFNA1. This second GWAS was also conducted using data from the UK Biobank and employed the Olink proteomic platform; however, it identified a different lead genetic variant associated with EFNA1.

The results from MR sensitivity analyses, employing these alternative genetic instruments for EFNA1, yielded effect estimates for the association between genetically predicted circulating EFNA1 levels and forearm fracture risk that were largely consistent with that reported for the genetic instrument used in the present study (Table S6). To our knowledge, no genetic instruments for EFNA1 have been identified using other proteomics platforms. Unfortunately, extensive MR sensitivity analyses (e.g., Egger MR, weighted median MR, leave-one-out MR) are not feasible when only one independent cis-pQTL is available.

In addition, we assessed the association between genetically predicted circulating levels of EFNA1 and risk of fracture at any skeletal site.³ This analysis revealed a statistically significant association in the same direction as that observed for forearm fractures (Table S6).

We have added the following section to Results (Page 7, Line 144)

“As EFNA1 was selected as a promising candidate for our further downstream analyses, we first conducted MR sensitivity analyses using alternative genetic instruments for EFNA1. The results from these MR sensitivity analyses, employing alternative genetic instruments for EFNA1, yielded effect estimates for the association between genetically predicted circulating EFNA1 levels and forearm fracture risk that were largely consistent with that reported in the present study (Table S6).”

In addition, we assessed the association between genetically predicted circulating levels of EFNA1 and risk of fracture at any skeletal site.³ This analysis revealed a statistically significant association in the same direction as that observed for forearm fractures (Table S6).”

Table S6 Mendelian randomization (MR) sensitivity analyses of the association between genetically determined circulating EFNA1 and fracture risk using alternative genetic instruments and outcomes

Outcome	IV for exposure	OR	95% CI	P	Exposure source
Forearm fracture					
	rs7546746	0.90	0.85 0.94	4.9E-06	PMID: 37794188
	rs4745	0.90	0.86 0.94	2.2E-06	PMID: 37794188
	rs4390169	0.90	0.86 0.94	2.5E-06	PMID: 37794186
Any fracture					
	rs7546746	0.96	0.92 1.00	4.1E-02	PMID: 37794188
	rs4745	NA	NA NA	NA	PMID: 37794188
	rs4390169	0.95	0.92 1.00	2.9E-02	PMID: 37794186

First, we used a different cis-pQTL (rs4745, instead of the initially used SNP rs7546746) identified in the same GWAS of circulating EFNA1 based on the Olink proteomic platform (PMID: 37794188). This variant was initially excluded from the primary analyses due to linkage disequilibrium ($R^2 > 0.01$) with rs7546746. Second, we utilized another genetic instrument (rs4390169) derived from a separate large-scale proteome-wide GWAS (PMID: 37794186), which also included EFNA1. This second GWAS was also conducted using data from the UK Biobank and employed the Olink proteomic platform; however, it identified a different lead genetic variant associated with EFNA1. In addition, we assessed the association between genetically predicted circulating levels of EFNA1 and risk of fracture at any skeletal site (PMID: 30598549). For the MR, Wald Ratio was calculated. NA = the SNP was not available in the used outcome GWAS summary statistics.

4. To better contextualize these findings within current research, could the authors provide more detailed insights into recent developments regarding ephrin-A1-EphA2 signaling as a target for fracture prevention?

AU: We have added the following new sections to the discussion:

Discussion (Page 15, Line 365)

“To the best of our knowledge, no prior studies have investigated ephrin-A1–EphA2 signaling as a potential therapeutic target for fracture prevention. However, previous in vitro studies have indicated a bidirectional interaction between osteoclastic ephrin-A2 and osteoblastic EphA2, promoting osteoclastogenesis while concurrently suppressing osteoblastic bone formation.⁴ The role of ephrin-B2–EphB4 signaling in bone homeostasis is more firmly established.⁴ Experimental evidence has shown that the interaction between osteoclastic ephrin-B2 and osteoblastic EphB4 suppresses bone resorption while promoting osteoblastic bone formation.⁴ Furthermore, in vivo studies by Bikle et al. have demonstrated that ephrin-B2–EphB4 signaling mediates the effects of insulin-like growth factor I (IGF-I) in the regulation of endochondral bone formation.⁵ The same research group also demonstrated that ephrin-B2 expression in Col2-positive cells plays a critical role in regulating fracture repair.⁶ Thus, although other ephrins have been implicated in bone homeostasis and fracture repair in experimental settings, the present human MR-colocalization pipeline identifies ephrin-A1 as

a key regulator of fracture risk in humans. Future studies are warranted to elucidate the potential role of ephrin-A1–EphA2 signaling in the regulation of fracture repair.

The data presented in this study suggest that membrane-bound ephrin-A1 expressed on vascular endothelial cells may directly interact with the EphA2 receptor on osteoblasts. We hypothesize that this ephrin-A1–EphA2 signaling promotes osteoblast-mediated bone formation, thereby contributing to increased bone mineral density. The drug targets underlying currently approved osteoporosis treatments were primarily identified as mechanisms with an effect on BMD.⁷ Importantly, the heritable component of fracture risk is partly independent of BMD.^{8,9} It is therefore plausible that key druggable targets for fracture prevention have remained undiscovered due to prior focus on mechanisms affecting BMD. This notion is further supported by our recent large-scale GWAS on forearm fractures, which identified several genetic loci associated with bone quality parameters that were not detected in previous GWAS focused on BMD.⁷ Given that the current MR-colocalization pipeline utilizes the same forearm fracture GWAS for outcome analyses,⁷ we hypothesize that this approach enables the identification of both BMD-dependent and BMD-independent mechanisms contributing to fracture risk. Although we propose that a portion of the effect of ephrin-A1 on fracture risk may be mediated through alterations in BMD, the underlying mechanism for ephrin-A1 on fracture risk may also involve yet unidentified influences on bone quality.

Our comprehensive descriptive analyses revealed a marked reduction in the expression of ephrin-A1 and/or its high-affinity receptor EphA2 in bone tissue in murine models of bone loss induced by inflammation, aging, or high-dose vitamin A administration. In contrast, their expression remained unchanged in a model of bone loss induced by ovariectomy. Moreover, the anabolic bone response elicited by mechanical loading, but not by intermittent parathyroid hormone (PTH) administration, was associated with upregulated expression of ephrin-A1 in bone tissue. The observed pattern of altered ephrin-A1 and EphA2 expression in selected osteoporosis disease models, as well as in response to certain anabolic stimuli, suggests that ephrin-A1–EphA2 signaling in bone may represent a mechanistically distinct pathway from those targeted by currently approved osteoporosis therapies. Based on these findings, we hypothesize that therapeutic strategies targeting ephrin-A1–EphA2 signaling may act synergistically with existing antifracture treatments, offering complementary and potentially additive benefits in the management of osteoporosis. Furthermore, therapeutic modulation of ephrin-A1–EphA2 signaling may not be associated with the increased risk of serious, albeit rare, adverse events linked to long-term use of currently approved osteoporosis medications.^{10,11} This could potentially enhance patient adherence to treatment.”

5. A minor issue: inconsistent notation: "EphA2" and "Epha2" are used in the manuscript.

AU: Now corrected to always be EphA2.

Reviewer #2 (Remarks to the Author):

What are the noteworthy results?

Based on a large genetic patient screening of osteoporotic forearm fracture cases the author group presents several known genes that are currently clinically targeted by approved drugs (sclerostin and osteoprotegerin) and three other proteins (DKK1, RSPO3, and MEPE) are well established bone-related proteins and a group of three so far unknown genes that were now analyzed in more detail. In subsequent mechanistic studies, they focused on one of these novel proteins, ephrin-A1, they found to be specifically associated with forearm fractures.

AU: We agree with this description.

Will the work be of significance to the field and related fields?

Conceptually, the work presents on the one hand an interesting approach to identify so far unknown novel targets. These targets may indeed be interesting to follow and understand their role in more depth and compare to the established understanding of osteoporosis. On the other hand, the work fails to bring the novel target in a relationship to bone homeostasis on a more general level or on the specific level of known derailed homeostasis in osteoporosis.

AU: We have included new sections in the discussion highlighting the potential relevance of our findings to the field of osteoporosis.

Discussion (Page 15, Line 365)

“To the best of our knowledge, no prior studies have investigated ephrin-A1–EphA2 signaling as a potential therapeutic target for fracture prevention. However, previous in vitro studies have indicated a bidirectional interaction between osteoclastic ephrin-A2 and osteoblastic EphA2, promoting osteoclastogenesis while concurrently suppressing osteoblastic bone formation.⁴ The role of ephrin-B2–EphB4 signaling in bone homeostasis is more firmly established.⁴ Experimental evidence has shown that the interaction between osteoclastic ephrin-B2 and osteoblastic EphB4 suppresses bone resorption while promoting osteoblastic bone formation.⁴ Furthermore, in vivo studies by Bikle et al. have demonstrated that ephrin-B2–EphB4 signaling mediates the effects of insulin-like growth factor I (IGF-I) in the regulation of endochondral bone formation.⁵ The same research group also demonstrated that ephrin-B2 expression in Col2-positive cells plays a critical role in regulating fracture repair.⁶ Thus, although other ephrins have been implicated in bone homeostasis and fracture repair in experimental settings, the present human MR-colocalization pipeline identifies ephrin-A1 as a key regulator of fracture risk in humans. Future studies are warranted to elucidate the potential role of ephrin-A1–EphA2 signaling in the regulation of fracture repair.

The data presented in this study suggest that membrane-bound ephrin-A1 expressed on vascular endothelial cells may directly interact with the EphA2 receptor on osteoblasts. We hypothesize that this ephrin-A1–EphA2 signaling promotes osteoblast-mediated bone formation, thereby contributing to increased bone mineral density. The drug targets underlying currently approved osteoporosis treatments were primarily identified as mechanisms with an effect on BMD.⁷ Importantly, the heritable component of fracture risk is partly independent of BMD.^{8,9} It is therefore plausible that key druggable targets for fracture prevention have remained undiscovered due to prior focus on mechanisms affecting BMD. This notion is further supported by our recent large-scale GWAS on forearm fractures, which identified several genetic loci associated with bone quality parameters that were not detected in previous GWAS focused on BMD.⁷ Given that the current MR-colocalization pipeline utilizes the same

forearm fracture GWAS for outcome analyses,⁷ we hypothesize that this approach enables the identification of both BMD-dependent and BMD-independent mechanisms contributing to fracture risk. Although we propose that a portion of the effect of ephrin-A1 on fracture risk may be mediated through alterations in BMD, the underlying mechanism for ephrin-A1 on fracture risk may also involve yet unidentified influences on bone quality.”

AU: In addition, we have performed extensive new analyses of the expressions of ephrin-A1 and its high-affinity receptor EphA2 in bone tissue in several disease models of bone loss, including those induced by inflammation, aging, high-dose vitamin A administration, and estrogen deficiency (New Figure 6A-D). Moreover, we have examined whether ephrin-A1 or EphA2 levels in bone tissue are affected by anabolic bone stimuli, specifically mechanical loading and intermittent PTH treatment (New Figure 6E, F). The observed patterns of altered ephrin-A1 and EphA2 expression in osteoporosis disease models, as well as in response to anabolic stimuli, suggest that ephrin-A1–EphA2 signaling in bone may represent a mechanistically distinct pathway, separate from those targeted by currently approved osteoporosis therapies.

Methods (Page 18, Line 474)

Mouse models for bone loss and anabolic bone formation

To investigate whether EfnA1 and/or EphA2 expression in bone tissue is regulated in osteoporosis disease models or by bone anabolic treatments, a variety of mouse models were used:

A systemic inflammation-induced bone loss model: *To induce systemic inflammation, 9-week-old female C57BL/6N mice were intravenously inoculated via the tail vein with 0.2 ml of Staphylococcus aureus (SA, 5×10^7 CFU/mouse) or PBS as vehicle control (Veh). Eight days post-inoculation, the mice were euthanized, and the vertebrae were dissected. Microcomputer tomography analysis revealed a significant reduction in bone volume over total volume (BV/TV) in the L5 vertebra of SA-treated mice ($-29.5 \pm 2.4\%$ compared to vehicle controls; Student's *t* test, $p = 1.9 \times 10^{-7}$).*

An age-induced bone loss model: *15-month-old and 2-month-old female C57BL/6N mice from the same strain (C57BL/6JRj) and breeder (Janvier Labs, France) were euthanized and vertebrae were dissected. A recent publication from our group demonstrates a significant reduction of trabecular bone volume fraction following aging.¹²*

A vitamin A-induced bone loss model: *Nine-week-old female C57BL/6N mice were fed either a control chow diet (Control) or a chow supplemented with a high dose of retinyl acetate (vitamin A, 700 IU/g; Teklad Custom Diets, Madison, WI). After four days of dietary intervention, the mice were euthanized, and tibia and femur were dissected. Peripheral quantitative computed tomography analysis of femur revealed a significant reduction in the cortical thickness following high dose vitamin A supplementation ($-6.0 \pm 1.7\%$ compared to control fed mice; Student's *t* test, $p = 1.7 \times 10^{-2}$).*

A postmenopausal osteoporosis model: *13-week-old female mice underwent ovariectomy or sham surgery. Four weeks post-surgery, the mice were euthanized, and tibia and femur were dissected. Peripheral quantitative computed tomography analysis of femur revealed a significant reduction in the cortical thickness following ovariectomy ($-7.0 \pm 1.5\%$ compared to sham mice; Student's *t* test, $p = 3.7 \times 10^{-4}$). Some data from these mice have been published previously (unrelated to the present study).¹³*

A loading-induced bone formation model: For 3 days a week for 2 weeks, the right tibia of 13-week-old female mice was subjected to loading by placing the tibia in the holding cups of the ElectroForce 3100 Test Instrument, under the sedation of isoflurane (Baxter). Axial load of 14.4 N was applied through the knee joint for 40 cycles with 10 s rest between the cycles. The left unloaded tibia served as control. As previously demonstrated, loading increased cortical thickness ($+25.2 \pm 1.6$ % compared to the non-loaded bone; Student's *t* test, $p = 9.3 \times 10^{-10}$).¹⁴

A PTH-induced bone formation model: 11-week-old male mice were treated with either vehicle (Veh, 20 mM NaH₂PO₄ in NaCl) or human parathyroid hormone 1-34 (PTH, dissolved in 20 mM NaH₂PO₄ in NaCl, Bachem). Treatments were administered via intraperitoneal injection 5 days a week for 3 weeks. Following the treatment period, mice were euthanized, and tibiae were dissected. Peripheral quantitative computed tomography analysis of tibia revealed a significant increase in cortical thickness in PTH-treated mice ($+11.7 \pm 1.6$ % compared to vehicle-treated control mice; Student's *t* test, $p = 6.2 \times 10^{-3}$).

Results (Page 9, Line 216)

“Finally, we investigated whether the expression of ephrin-A1 or its high-affinity receptor EphA2 in bone tissue are regulated across various osteoporosis models, including those with bone loss induced by inflammation, aging, high-dose vitamin A exposure, or estrogen deficiency (Fig. 6). Inflammation-induced bone loss was associated with decreased expression of both ephrin-A1 and EphA2 in bone tissue (Fig. 6A). Bone loss associated with aging was linked to reduced levels of EphA2 and a non-significant trend toward decreased expression of ephrin-A1 in bone tissue (Fig. 6B). Bone loss induced by high-dose vitamin A was specifically associated with reduced expression of EphA2 in bone tissue (Fig. 6C). Ovariectomy-induced bone loss, which models postmenopausal osteoporosis, was not associated with altered expression of ephrin-A1 or its receptor EphA2 in bone tissue (Fig. 6D). The anabolic bone response induced by mechanical loading, but not by intermittent administration of parathyroid hormone (PTH), was associated with increased expression of ephrin-A1 in bone tissue (Fig. 6E, F). The observed alterations in ephrin-A1 and EphA2 expression was dependent on the osteoporosis models, as well as on the type of anabolic stimuli evaluated. This suggests that ephrin-A1–EphA2 signaling in bone may constitute a mechanistically distinct pathway from those targeted by currently approved osteoporosis therapies. Nonetheless, further functional studies are needed to determine whether ephrin-A1–EphA2 signaling contributes to the observed bone effects in selected osteoporosis disease models or in the response to certain anabolic stimuli on bone mass.”

Discussion (Page 16, Line 397)

“Our comprehensive descriptive analyses revealed a marked reduction in the expression of ephrin-A1 and/or its high-affinity receptor EphA2 in bone tissue in murine models of bone loss induced by inflammation, aging, or high-dose vitamin A administration. In contrast, their expression remained unchanged in a model of bone loss induced by ovariectomy. Moreover, the anabolic bone response elicited by mechanical loading, but not by intermittent parathyroid hormone (PTH) administration, was associated with upregulated expression of ephrin-A1 in bone tissue. The observed patterns of altered ephrin-A1 and EphA2 expression in selected osteoporosis disease models, as well as in response to certain anabolic stimuli, suggest that ephrin-A1–EphA2 signaling in bone may represent a mechanistically distinct pathway from those targeted by currently approved osteoporosis therapies.

Based on these findings, we hypothesize that therapeutic strategies targeting ephrin-A1–EphA2 signaling may act synergistically with existing antifracture treatments, offering complementary and potentially additive benefits in the management of osteoporosis. Furthermore, therapeutic modulation of ephrin-A1–EphA2 signaling may not be associated with the increased risk of serious, albeit rare, adverse events linked to long-term use of currently approved osteoporosis medications.^{10,11} This could potentially enhance patient adherence to treatment. ”

The link between pre-osteoblasts and endothelial cells and their direct communication on the bone marrow niche and on the endothelial side is not novel and has been earlier described by Adams et al and others. How would this interaction be derailed in osteoporotic patient settings and how would such derailed setting impact bone mineral density or bone mass? Would it directly impact osteoblasts bone formation capacity or the indirect link through an increased osteoclasts activity? Also, would this novel path be competing or complementing the (known) osteoporotic derailments that current treatments aim to address.

AU: While the interaction between pre-osteoblasts and endothelial cells is well established, the specific role of EfnA1-Epha2 signaling in this coupling has not been previously reported. We have included new sections to the discussion highlighting the potential relevance of this interaction in the context of osteoporosis and regulation of BMD.

AU: We have added the following new sections to the discussion:

Discussion (Page 15, Line 365)

“To the best of our knowledge, no prior studies have investigated ephrin-A1–EphA2 signaling as a potential therapeutic target for fracture prevention. However, previous in vitro studies have indicated a bidirectional interaction between osteoclastic ephrin-A2 and osteoblastic EphA2, promoting osteoclastogenesis while concurrently suppressing osteoblastic bone formation.⁴ The role of ephrin-B2–EphB4 signaling in bone homeostasis is more firmly established.⁴ Experimental evidence has shown that the interaction between osteoclastic ephrin-B2 and osteoblastic EphB4 suppresses bone resorption while promoting osteoblastic bone formation.⁴ Furthermore, in vivo studies by Bikle et al. have demonstrated that ephrin-B2–EphB4 signaling mediates the effects of insulin-like growth factor I (IGF-I) in the regulation of endochondral bone formation.⁵ The same research group also demonstrated that ephrin-B2 expression in Col2-positive cells plays a critical role in regulating fracture repair.⁶ Thus, although other ephrins have been implicated in bone homeostasis and fracture repair in experimental settings, the present human MR-colocalization pipeline identifies ephrin-A1 as a key regulator of fracture risk in humans. Future studies are warranted to elucidate the potential role of ephrin-A1–EphA2 signaling in the regulation of fracture repair.

The data presented in this study suggest that membrane-bound ephrin-A1 expressed on vascular endothelial cells may directly interact with the EphA2 receptor on osteoblasts. We hypothesize that this ephrin-A1–EphA2 signaling promotes osteoblast-mediated bone formation, thereby contributing to increased bone mineral density. The drug targets underlying currently approved osteoporosis treatments were primarily identified as mechanisms with an effect on BMD.⁷ Importantly, the heritable component of fracture risk is partly independent of BMD.^{8,9} It is therefore plausible that key druggable

targets for fracture prevention have remained undiscovered due to prior focus on mechanisms affecting BMD. This notion is further supported by our recent large-scale GWAS on forearm fractures, which identified several genetic loci associated with bone quality parameters that were not detected in previous GWAS focused on BMD.⁷ Given that the current MR-colocalization pipeline utilizes the same forearm fracture GWAS for outcome analyses,⁷ we hypothesize that this approach enables the identification of both BMD-dependent and BMD-independent mechanisms contributing to fracture risk. Although we propose that a portion of the effect of ephrin-A1 on fracture risk may be mediated through alterations in BMD, the underlying mechanism for ephrin-A1 on fracture risk may also involve yet unidentified influences on bone quality.

Our comprehensive descriptive analyses revealed a marked reduction in the expression of ephrin-A1 and/or its high-affinity receptor EphA2 in bone tissue in murine models of bone loss induced by inflammation, aging, or high-dose vitamin A administration. In contrast, their expression remained unchanged in a model of bone loss induced by ovariectomy. Moreover, the anabolic bone response elicited by mechanical loading, but not by intermittent parathyroid hormone (PTH) administration, was associated with upregulated expression of ephrin-A1 in bone tissue. The observed pattern of altered ephrin-A1 and EphA2 expression in selected osteoporosis disease models, as well as in response to certain anabolic stimuli, suggests that ephrin-A1–EphA2 signaling in bone may represent a mechanistically distinct pathway from those targeted by currently approved osteoporosis therapies. Based on these findings, we hypothesize that therapeutic strategies targeting ephrin-A1–EphA2 signaling may act synergistically with existing antifracture treatments, offering complementary and potentially additive benefits in the management of osteoporosis. Furthermore, therapeutic modulation of ephrin-A1–EphA2 signaling may not be associated with the increased risk of serious, albeit rare, adverse events linked to long-term use of currently approved osteoporosis medications.^{10,11} This could potentially enhance patient adherence to treatment.”

Does the work support the conclusions and claims, or is additional evidence needed?

I am struggling with the claim of prevention that the ephrin-A1 should have. Prevention is very hard to show and prove. I would suggest putting this novel target (ephrin-A1) in relation to the known and druggable targets... I would strongly suggest to re-consider the current manuscript punch line and relate the work to established osteoporotic strategies and concepts and thereby open the possibility of a novel perspective.

AU: We have included new sections in the discussion that aim to connect our findings with established osteoporotic strategies and concepts, thereby introducing the possibility of a novel perspective.

Discussion (Page 15, Line 365)

“To the best of our knowledge, no prior studies have investigated ephrin-A1–EphA2 signaling as a potential therapeutic target for fracture prevention. However, previous in vitro studies have indicated a bidirectional interaction between osteoclastic ephrin-A2 and osteoblastic EphA2, promoting osteoclastogenesis while concurrently suppressing osteoblastic bone formation.⁴ The role of ephrin-B2–EphB4 signaling in bone homeostasis is more firmly established.⁴ Experimental evidence has shown that the interaction between osteoclastic ephrin-B2 and osteoblastic EphB4 suppresses bone resorption while promoting osteoblastic bone formation.⁴ Furthermore, in vivo studies by Bikle et al. have demonstrated that ephrin-B2–EphB4 signaling mediates the effects of insulin-like growth factor I (IGF-I) in the regulation of endochondral bone formation.⁵ The same research group also

demonstrated that ephrin-B2 expression in Col2-positive cells plays a critical role in regulating fracture repair.⁶ Thus, although other ephrins have been implicated in bone homeostasis and fracture repair in experimental settings, the present human MR-colocalization pipeline identifies ephrin-A1 as a key regulator of fracture risk in humans. Future studies are warranted to elucidate the potential role of ephrin-A1–EphA2 signaling in the regulation of fracture repair.

The data presented in this study suggest that membrane-bound ephrin-A1 expressed on vascular endothelial cells may directly interact with the EphA2 receptor on osteoblasts. We hypothesize that this ephrin-A1–EphA2 signaling promotes osteoblast-mediated bone formation, thereby contributing to increased bone mineral density. The drug targets underlying currently approved osteoporosis treatments were primarily identified as mechanisms with an effect on BMD.⁷ Importantly, the heritable component of fracture risk is partly independent of BMD.^{8,9} It is therefore plausible that key druggable targets for fracture prevention have remained undiscovered due to prior focus on mechanisms affecting BMD. This notion is further supported by our recent large-scale GWAS on forearm fractures, which identified several genetic loci associated with bone quality parameters that were not detected in previous GWAS focused on BMD.⁷ Given that the current MR-colocalization pipeline utilizes the same forearm fracture GWAS for outcome analyses,⁷ we hypothesize that this approach enables the identification of both BMD-dependent and BMD-independent mechanisms contributing to fracture risk. Although we propose that a portion of the effect of ephrin-A1 on fracture risk may be mediated through alterations in BMD, the underlying mechanism for ephrin-A1 on fracture risk may also involve yet unidentified influences on bone quality.

Our comprehensive descriptive analyses revealed a marked reduction in the expression of ephrin-A1 and/or its high-affinity receptor EphA2 in bone tissue in murine models of bone loss induced by inflammation, aging, or high-dose vitamin A administration. In contrast, their expression remained unchanged in a model of bone loss induced by ovariectomy. Moreover, the anabolic bone response elicited by mechanical loading, but not by intermittent parathyroid hormone (PTH) administration, was associated with upregulated expression of ephrin-A1 in bone tissue. The observed pattern of altered ephrin-A1 and EphA2 expression in selected osteoporosis disease models, as well as in response to certain anabolic stimuli, suggests that ephrin-A1–EphA2 signaling in bone may represent a mechanistically distinct pathway from those targeted by currently approved osteoporosis therapies. Based on these findings, we hypothesize that therapeutic strategies targeting ephrin-A1–EphA2 signaling may act synergistically with existing antifracture treatments, offering complementary and potentially additive benefits in the management of osteoporosis. Furthermore, therapeutic modulation of ephrin-A1–EphA2 signaling may not be associated with the increased risk of serious, albeit rare, adverse events linked to long-term use of currently approved osteoporosis medications.^{10,11} This could potentially enhance patient adherence to treatment.”

Reviewer #3 (Remarks to the Author):

This article presents a compelling investigation of circulating proteins associated with fracture risk using MR and colocalization analysis. A significant highlight is the novel 3D DeepBone tissue-clearing technique which allows for detailed spatial analysis of EFNA1 and EPHA2 localization in bone tissue. The findings regarding EFNA1's interaction with EPHA2 provide valuable insights into the mechanisms underlying bone fractures and their prevention.

AU: We sincerely appreciate these positive comments.

Suggestions for Improvement

1. While the authors suggest that EFNA1 may primarily be expressed by vascular endothelial cells, there is a lack of direct experimental evidence to support this claim. I recommend adding experiments that include overexpressing EFNA1 in endothelial cells and assessing its effects on osteoblast lineage differentiation and mineralization. Additionally, performing conditional knockout of the EFNA1 gene in vascular endothelial cells to observe the impact on bone mass would be beneficial.

AU: We fully agree that substantial additional work including functional studies employing novel mouse models with conditional inactivation of EFNA1 in endothelial cells and other possible candidate cells, as well as models with endothelial-specific overexpression of EFNA1, will be required to determine the cellular origin of EFNA1 with an impact on bone health. These experiments are extensive and would require an additional 2 years of animal and laboratory work, including the development of several novel mouse models. We believe that the comprehensive MR-colocalization pipeline employed in this study, combined with detailed expression analyses using single-cell RNA sequencing and novel spatial transcriptomics, as well as functional validation in mouse models with global gene inactivation, collectively provides sufficient scientific merit to warrant publication without the inclusion of additional in-depth functional investigations. This notion is further supported by Reviewer 3's fourth comment, stating: 'Overall, this is an excellent study that opens new avenues for research on EFNA1 and its role in bone health.' Furthermore, we believe that promptly reporting this human-related observation is crucial, as it may enhance ongoing scientific efforts to combat osteoporosis and encourage further investigation into this potentially novel and highly human-relevant mechanism of bone homeostasis.

However, as described below, we have now clearly stated that these suggested additional functional studies are required to determine the origin of EFNA1 with an impact on bone health.

Discussion (Page 14, Line 355)

“To further elucidate the cellular origin of EFNA1 relevant to bone health, future functional studies employing novel mouse models with conditional inactivation of EFNA1 in endothelial cells and other candidate cells, as well as models with endothelial-specific overexpression of EFNA1, will be required.”

2. Including a longitudinal assessment of EFNA1 and EphA2 levels in various fracture risk models would be advantageous. This approach could help clarify whether elevated levels of these proteins serve as biomarkers for increased fracture risk over time.

AU: We have performed extensive new analyses of the expressions of ephrin-A1 and its high-affinity receptor EphA2 in bone tissue in several disease models of bone loss, including those induced by inflammation, aging, high-dose vitamin A administration, and estrogen deficiency (New Figure 6A-D). Moreover, we have examined whether ephrin-A1 or EphA2 levels in bone tissue are affected by anabolic bone stimuli, specifically mechanical loading and intermittent PTH treatment (New Figure 6E, F). The observed patterns of altered ephrin-A1 and EphA2 expression in osteoporosis disease models, as well as in response to anabolic stimuli, suggest that ephrin-A1–

EphA2 signaling in bone may represent a mechanistically distinct pathway, separate from those targeted by currently approved osteoporosis therapies.

Methods (Page 18, Line 474)

Mouse models for bone loss and anabolic bone formation

To investigate whether Efn1 and/or EphA2 expression in bone tissue is regulated in osteoporosis disease models or by bone anabolic treatments, a variety of mouse models were used:

*A systemic inflammation-induced bone loss model: To induce systemic inflammation, 9-week-old female C57BL/6N mice were intravenously inoculated via the tail vein with 0.2 ml of *Staphylococcus aureus* (SA, 5×10^7 CFU/mouse) or PBS as vehicle control (Veh). Eight days post-inoculation, the mice were euthanized, and the vertebrae were dissected. Microcomputer tomography analysis revealed a significant reduction in bone volume over total volume (BV/TV) in the L5 vertebra of SA-treated mice ($-29.5 \pm 2.4\%$ compared to vehicle controls; Student's *t* test, $p = 1.9 \times 10^{-7}$).*

An age-induced bone loss model: 15-month-old and 2-month-old female C57BL/6N mice from the same strain (C57BL/6JRj) and breeder (Janvier Labs, France) were euthanized and vertebrae were dissected. A recent publication from our group demonstrates a significant reduction of trabecular bone volume fraction following aging.¹²

*A vitamin A-induced bone loss model: Nine-week-old female C57BL/6N mice were fed either a control chow diet (Control) or a chow supplemented with a high dose of retinyl acetate (vitamin A, 700 IU/g) (Teklad Custom Diets, Madison, WI). After four days of dietary intervention, the mice were euthanized, and tibia and femur were dissected. Peripheral quantitative computed tomography analysis of femur revealed a significant reduction in the cortical thickness following high dose vitamin A supplementation ($-6.0 \pm 1.7\%$ compared to control fed mice; Student's *t* test, $p = 1.7 \times 10^{-2}$).*

*A postmenopausal osteoporosis model: 13-week-old female mice underwent ovariectomy or sham surgery. Four weeks post-surgery, the mice were euthanized, and tibia and femur were dissected. Peripheral quantitative computed tomography analysis of femur revealed a significant reduction in the cortical thickness following ovariectomy ($-7.0 \pm 1.5\%$ compared to sham mice; Student's *t* test, $p = 3.7 \times 10^{-4}$). Some data from these mice have been published previously (unrelated to the present study).¹³*

*A loading-induced bone formation model: For 3 days a week for 2 weeks, the right tibia of 13-week-old female mice was subjected to loading by placing the tibia in the holding cups of the ElectroForce 3100 Test Instrument, under the sedation of isoflurane (Baxter). Axial load of 14.4 N was applied through the knee joint for 40 cycles with 10 s rest between the cycles. The left unloaded tibia served as control. As previously demonstrated, loading increased cortical thickness ($+25.2 \pm 1.6\%$ compared to the non-loaded bone; Student's *t* test, $p = 9.3 \times 10^{-10}$).¹⁴*

*A PTH-induced bone formation model: 11-week-old male mice were treated with either vehicle (Veh, 20 mM NaH₂PO₄ in NaCl) or human parathyroid hormone 1-34 (PTH, dissolved in 20 mM NaH₂PO₄ in NaCl, Bachem). Treatments were administered via intraperitoneal injection 5 days a week for 3 weeks. Following the treatment period, mice were euthanized, and tibiae were dissected. Peripheral quantitative computed tomography analysis of tibia revealed a significant increase in cortical thickness in PTH-treated mice ($+11.7 \pm 1.6\%$ compared to vehicle-treated control mice; Student's *t* test, $p = 6.2 \times 10^{-3}$).*

Results (Page 9, Line 216)

“Finally, we investigated whether the expression of ephrin-A1 or its high-affinity receptor EphA2 in bone tissue are regulated across various osteoporosis models, including those with bone loss induced by inflammation, aging, high-dose vitamin A exposure, or estrogen deficiency (Fig. 6). Inflammation-induced bone loss was associated with decreased expression of both ephrin-A1 and EphA2 in bone tissue (Fig. 6A). Bone loss associated with aging was linked to reduced levels of EphA2 and a non-significant trend toward decreased expression of ephrin-A1 in bone tissue (Fig. 6B). Bone loss induced by high-dose vitamin A was specifically associated with reduced expression of EphA2 in bone tissue (Fig. 6C). Ovariectomy-induced bone loss, which models postmenopausal osteoporosis, was not associated with altered expression of ephrin-A1 or its receptor EphA2 in bone tissue (Fig. 6D). The anabolic bone response induced by mechanical loading, but not by intermittent administration of parathyroid hormone (PTH), was associated with increased expression of ephrin-A1 in bone tissue (Fig. 6E, F). The observed alterations in ephrin-A1 and EphA2 expression was dependent on the osteoporosis models, as well as on the type of anabolic stimuli evaluated. This suggests that ephrin-A1–EphA2 signaling in bone may constitute a mechanistically distinct pathway from those targeted by currently approved osteoporosis therapies. Nonetheless, further functional studies are needed to determine whether ephrin-A1–EphA2 signaling contributes to the observed bone effects in selected osteoporosis disease models or in the response to certain anabolic stimuli on bone mass.”

Discussion (Page 16, Line 397)

“Our comprehensive descriptive analyses revealed a marked reduction in the expression of ephrin-A1 and/or its high-affinity receptor EphA2 in bone tissue in murine models of bone loss induced by inflammation, aging, or high-dose vitamin A administration. In contrast, their expression remained unchanged in a model of bone loss induced by ovariectomy. Moreover, the anabolic bone response elicited by mechanical loading, but not by intermittent parathyroid hormone (PTH) administration, was associated with upregulated expression of ephrin-A1 in bone tissue. The observed patterns of altered ephrin-A1 and EphA2 expression in selected osteoporosis disease models, as well as in response to certain anabolic stimuli, suggest that ephrin-A1–EphA2 signaling in bone may represent a mechanistically distinct pathway from those targeted by currently approved osteoporosis therapies. Based on these findings, we hypothesize that therapeutic strategies targeting ephrin-A1–EphA2 signaling may act synergistically with existing antifracture treatments, offering complementary and potentially additive benefits in the management of osteoporosis. Furthermore, therapeutic modulation of ephrin-A1–EphA2 signaling may not be associated with the increased risk of serious, albeit rare, adverse events linked to long-term use of currently approved osteoporosis medications.^{10,11} This could potentially enhance patient adherence to treatment. ”

3. The authors should also consider discussing the clinical implications of their findings, such as potential therapeutic targets for fracture prevention. Exploring how modifying EFNA1 or EPHA2 expression might influence fracture healing or risk in clinical settings would enhance the relevance of this research.

AU: We have included new sections in the discussion highlighting the potential relevance of our findings to the field of osteoporosis.

Discussion (Page 15, Line 365)

“To the best of our knowledge, no prior studies have investigated ephrin-A1–EphA2 signaling as a potential therapeutic target for fracture prevention. However, previous in vitro studies have indicated a bidirectional interaction between osteoclastic ephrin-A2 and osteoblastic EphA2, promoting osteoclastogenesis while concurrently suppressing osteoblastic bone formation.⁴ The role of ephrin-B2–EphB4 signaling in bone homeostasis is more firmly established.⁴ Experimental evidence has shown that the interaction between osteoclastic ephrin-B2 and osteoblastic EphB4 suppresses bone resorption while promoting osteoblastic bone formation.⁴ Furthermore, in vivo studies by Bikle et al. have demonstrated that ephrin-B2–EphB4 signaling mediates the effects of insulin-like growth factor I (IGF-I) in the regulation of endochondral bone formation.⁵ The same research group also demonstrated that ephrin-B2 expression in Col2-positive cells plays a critical role in regulating fracture repair.⁶ Thus, although other ephrins have been implicated in bone homeostasis and fracture repair in experimental settings, the present MR-colocalization pipeline identifies ephrin-A1 as a key regulator of fracture risk in humans. Future studies are warranted to elucidate the potential role of ephrin-A1–EphA2 signaling in the regulation of fracture repair.

The data presented in this study suggest that membrane-bound ephrin-A1 expressed on vascular endothelial cells may directly interact with the EphA2 receptor on osteoblasts. We hypothesize that this ephrin-A1–EphA2 signaling promotes osteoblast-mediated bone formation, thereby contributing to increased bone mineral density. The drug targets underlying currently approved osteoporosis treatments were primarily identified as mechanisms with an effect on BMD.⁷ Importantly, the heritable component of fracture risk is partly independent of BMD.^{8,9} It is therefore plausible that key druggable targets for fracture prevention have remained undiscovered due to prior focus on mechanisms affecting BMD. This notion is further supported by our recent large-scale GWAS on forearm fractures, which identified several genetic loci associated with bone quality parameters that were not detected in previous GWAS focused on BMD.⁷ Given that the current MR-colocalization pipeline utilizes the same forearm fracture GWAS for outcome analyses,⁷ we hypothesize that this approach enables the identification of both BMD-dependent and BMD-independent mechanisms contributing to fracture risk. Although we propose that a portion of the effect of ephrin-A1 on fracture risk may be mediated through alterations in BMD, the underlying mechanism for ephrin-A1 on fracture risk may also involve yet unidentified influences on bone quality.

Our comprehensive descriptive analyses revealed a marked reduction in the expression of ephrin-A1 and/or its high-affinity receptor EphA2 in bone tissue in murine models of bone loss induced by inflammation, aging, or high-dose vitamin A administration. In contrast, their expression remained unchanged in a model of bone loss induced by ovariectomy. Moreover, the anabolic bone response elicited by mechanical loading, but not by intermittent parathyroid hormone (PTH) administration, was associated with upregulated expression of ephrin-A1 in bone tissue. The observed pattern of altered ephrin-A1 and EphA2 expression in selected osteoporosis disease models, as well as in response to certain anabolic stimuli, suggests that ephrin-A1–EphA2 signaling in bone may represent a mechanistically distinct pathway from those targeted by currently approved osteoporosis therapies. Based on these findings, we hypothesize that therapeutic strategies targeting ephrin-A1–EphA2 signaling may act synergistically with existing antifracture treatments, offering complementary and potentially additive benefits in the management of osteoporosis. Furthermore, therapeutic modulation of ephrin-A1–EphA2 signaling may not be associated with the increased risk of serious, albeit rare,

adverse events linked to long-term use of currently approved osteoporosis medications.^{10,11} This could potentially enhance patient adherence to treatment.”

4. Overall, this is an excellent study that opens new avenues for research on EFNA1 and its role in bone health.

AU: We sincerely appreciate these positive comments.

References

1. Eldjarn, G.H., *et al.* Large-scale plasma proteomics comparisons through genetics and disease associations. *Nature* **622**, 348-358 (2023).
2. Sun, B.B., *et al.* Plasma proteomic associations with genetics and health in the UK Biobank. *Nature* **622**, 329-338 (2023).
3. Morris, J.A., *et al.* An atlas of genetic influences on osteoporosis in humans and mice. *Nat Genet* **51**, 258-266 (2019).
4. Matsuo, K. & Otaki, N. Bone cell interactions through Eph/ephrin: bone modeling, remodeling and associated diseases. *Cell Adh Migr* **6**, 148-156 (2012).
5. Wang, Y., *et al.* Ephrin B2/EphB4 mediates the actions of IGF-I signaling in regulating endochondral bone formation. *J Bone Miner Res* **29**, 1900-1913 (2014).
6. Wang, Y., *et al.* Ablation of Ephrin B2 in Col2 Expressing Cells Delays Fracture Repair. *Endocrinology* **161**(2020).
7. Nethander, M., *et al.* An atlas of genetic determinants of forearm fracture. *Nat Genet* **55**, 1820-1830 (2023).
8. Andrew, T., Antoniadou, L., Scurrah, K.J., Macgregor, A.J. & Spector, T.D. Risk of wrist fracture in women is heritable and is influenced by genes that are largely independent of those influencing BMD. *J Bone Miner Res* **20**, 67-74 (2005).
9. Ralston, S.H. & Uitterlinden, A.G. Genetics of osteoporosis. *Endocr Rev* **31**, 629-662 (2010).
10. Anastasilakis, A.D., *et al.* Osteonecrosis of the Jaw and Antiresorptive Agents in Benign and Malignant Diseases: A Critical Review Organized by the ECTS. *J Clin Endocrinol Metab* **107**, 1441-1460 (2022).
11. Black, D.M., *et al.* Atypical Femur Fracture Risk versus Fragility Fracture Prevention with Bisphosphonates. *N Engl J Med* **383**, 743-753 (2020).
12. Lawenius, L., *et al.* Transplantation of gut microbiota from old mice into young healthy mice reduces lean mass but not bone mass. *Gut Microbes* **15**, 2236755 (2023).
13. Nilsson, K.H., *et al.* Estradiol and RSPO3 regulate vertebral trabecular bone mass independent of each other. *American journal of physiology. Endocrinology and metabolism* **322**, E211-e218 (2022).
14. Lionikaite, V., *et al.* Vitamin A decreases the anabolic bone response to mechanical loading by suppressing bone formation. *FASEB journal : official publication of the Federation of American Societies for Experimental Biology* **33**, 5237-5247 (2019).

A point-by-point response to the reviewers' comments

Reviewer #1 (Remarks to the Author):

Thank the authors for their efforts to address my comments.

AU: We are grateful for this comment

Reviewer #2 (Remarks to the Author):

thank you for addressing my concerns - I have no further comments!

AU: We are grateful for this comment

Reviewer #3 (Remarks to the Author):

I have re-reviewed the revised manuscript. The authors have thoroughly addressed the reviewers' comments and made substantive improvements to the manuscript. The detailed assessment of ephrin-A1/EphA2 expression across multiple bone-loss and bone-anabolic models (new Figure 6) is particularly valuable, and the authors have clearly outlined necessary future functional studies in the Discussion. While additional cell-type-specific functional validation would be desirable, the current evidence provides sufficient merit for publication. I recommend acceptance of the manuscript.

AU: We are grateful for this comment